# Antimicrobial peptide class that forms discrete β-barrel stable pores anchored by transmembrane helices

Seth W. Dickey [1,2,3] ✉, Dylan J. Burgin[1], Ama N. Antwi [2], Amer Villaruz[1], Madeline R. Galac[4], Gordon Y. C. Cheung [1], Tatiana K. Rostovtseva [5], Liam J. Worrall[6], Aleksander C. Lazarski [6], Elio A. Cino[7], D. Peter Tieleman [7], Sergey M. Bezrukov[5], Natalie C. J. Strynadka[6] & Michael Otto [1] ✉

Bacteriocin peptides are weapons of inter-bacterial warfare and belong to the larger group of antimicrobial peptides (AMPs), which are frequently proposed as alternatives to antibiotics. Many AMPs kill by destroying the target's cytoplasmic membrane using short-lived membrane perturbations. Contrastingly, protein toxins form large pores by stably assembling in the target membrane. Here we describe an AMP class termed TMcins (for *trans*membrane helix-containing bacterio*cin*), in which half of the AMP forms a transmembrane helix. This characteristic allows TMcin to assemble into stable and large oligomeric pores. The biosynthetic locus of TMcin, which was broadly active against Gram-positive bacteria, is distributed throughout two major bacterial phyla, yet bears no homology to previously reported bacteriocin biosynthetic gene clusters. Our discovery of an AMP class that achieves pore stability otherwise only found in protein toxins transforms our current understanding of AMP structure and function and underscores the continuing importance of phenotype-initiated investigations in uncovering wholly uncharacterized antimicrobials.

Antimicrobial peptides (AMPs) are produced within all domains of life[1]. They are often sought as alternatives to antibiotics, to which many important pathogens have developed resistance[2,3]. Historically, AMPs were discovered after phenotyping producing organisms for antimicrobial activity[4–6]. More recently, bioinformatic genome mining approaches have revealed additional AMPs using prior knowledge of AMPs and their biosynthetic gene clusters (BGCs), a method that has been most successfully applied for bacteriocins, which are AMPs that are produced by bacteria to kill microbial competitors[7–15]. However, because microbial genome annotations remain incomplete,

whole antimicrobial classes may go unrecognized, which conceals unknown mechanisms of action from further investigation.

AMPs frequently kill cells by permeabilizing membranes[16]. However, due to the small peptide size (<10 kDa), it is rare for an AMP to form a stable permeabilizing structure within membranes, for which current examples are limited to sub-nanometer proton or cation conducting channels[17,18]. Instead, a landscape of system-dependent dynamic conformations has been proposed to explain AMP-membrane interactions[19,20]. This stands in contrast to the much larger pore-forming proteins, such as for example staphylococcal alpha-toxin or

[1]National Institute of Allergy and Infectious Diseases, National Institutes of Health, Bethesda, MD, USA. [2]Department of Veterinary Medicine, University of Maryland, College Park, MD, USA. [3]Virginia-Maryland College of Veterinary Medicine, College Park, MD, USA. [4]Bioinformatics and Computational Biosciences Branch, Office of Cyber Infrastructure and Computational Biology, National Institute of Allergy and Infectious Diseases, National Institutes of Health, Bethesda, MD, USA. [5]Eunice Kennedy Shriver National Institute of Child Health and Human Development, National Institutes of Health, Bethesda, MD, USA. [6]Department of Biochemistry and Molecular Biology and the Centre for Blood Research, University of British Columbia, Vancouver, BC, Canada. [7]Department of Biological Sciences and Centre for Molecular Simulation, University of Calgary, Calgary, Canada. ✉e-mail: sdickey@umd.edu; motto@niaid.nih.gov

streptococcal pneumolysin, which form discrete pore structures held together through extensive bonding networks to create aqueous channels on the nanometer scale[21–24]. Additionally, AMPs and pore-forming proteins apply distinct solutions to the challenge of diffusing through aqueous spaces to then partition into a hydrophobic membrane: while smaller AMPs exhibit amphipathic properties to balance lipid-water phase partitioning, pore-forming proteins undergo conformational changes, shielding hydrophobic regions from water as soluble monomers before inserting them into membranes upon oligomerization[21,24,25].

Intrigued by the antimicrobial activity produced by a small cluster of *Staphylococcus aureus* strains, we have uncovered the *trans*membrane helix-containing bacterio*cin* (TMcin) family, an atypical family of AMPs that possesses a full transmembrane helix (TMH) and oligomerizes to form large aqueous pores. We further define the TMcin BGC, which includes a distinct combination of genes that encode for mostly membrane proteins and is widespread throughout Grampositive bacteria.

## Results

### Antimicrobial activity and biosynthetic gene cluster (BGC)

We observed diffusible antimicrobial activity that was specific to one cluster, CT545, among methicillin-resistant *Staphylococcus aureus*

(MRSA) sequence-type (ST) 88 strains isolated from Buruli ulcer wounds of patients in West Africa (Fig. 1a)[26,27]. Importantly, the genetically highly similar W13 outlier isolate[26], which encodes the same virulence factors[28], lacked antimicrobial production. We then searched the short-read genome sequencing contigs using antiSMASH but found nothing specific to the CT545 cluster: the same BGCs and the putative lactococcin 972 family bacteriocin[29] were also present in W13 (Supplementary Table 1)[30]. Manually inspecting the genomes[26], we noticed a 9.4 kb contig specific to and identical within the CT545 isolates. Further examination revealed plasmid replication and mobilization genes, indicating that the contig represents a circular plasmid, which we designated as pCT545 (Supplementary Fig. 1a, b). In addition to genes coding for a putative toxin/anti-toxin (TA), pCT545 encodes several putative accessory genes (Fig. 1b), the combination of which does not resemble any known BGC. We therefore hypothesized that the accessory genes constituted an undescribed BGC that confers biosynthetic capability of an unknown class of antimicrobials. To test our hypothesis, we cured the CT545 isolate G1905 of its native pCT545 plasmid by replacing it with an engineered plasmid encoding kanamycin resistance and the putative pCT545 plasmid replication and TA system genes (G1905 pCT545::pCURE, Supplementary Fig. 1c). The G1905 pCT545::pCURE strain failed to produce diffusible antimicrobial

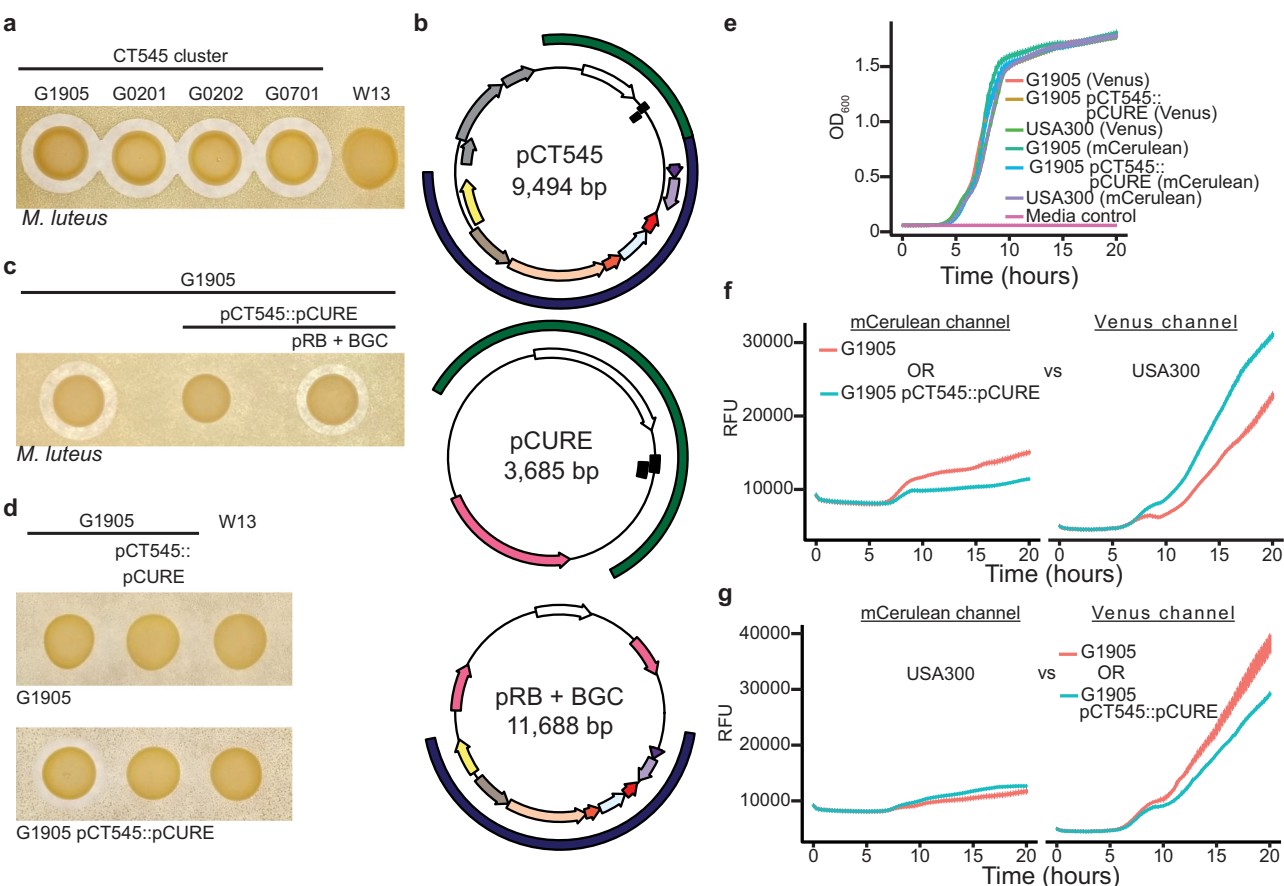

**Fig. 1 | Discovery of a novel bacteriocin AMP and BGC. a** Antimicrobial activity indicated by a zone of clearance is restricted to the CT545 *S. aureus* cluster. The sensitive indicator strain *Micrococcus luteus* (below-left) is embedded in agar and the test strains (above) spotted. **b** Plasmid maps of the native pCT545 (NZ_LFNJ01000005.1) and the engineered plasmids pCURE and pRB + BGC for pCT545 replacement (pCT545::pCURE) and BGC complementation, respectively. Outside arcs: plasmid replication and toxin-antitoxin system included in the pCURE plasmid (green) and BGC included in the complementation pRB+BGC plasmid (blue). Inside arcs: for BGC coloring, see Fig. 3b; toxin/antitoxin, black rectangles; putative phage mobilization, grey; replication initiator, white; antibiotic resistance

markers, pink. **c** The pCT545 BGC is necessary for antimicrobial activity and **d** an immunity mechanism is encoded by pCT545. **e** Monoculture growth measured by optical density at 600 nm (OD$_{600}$) of G1905 and USA300 strains expressing Venus or mCerulean fluorescent proteins. **f** Co-culture growth of G1905 or G1905 pCT545::pCURE expressing mCerulean vs USA300 expressing Venus. Left: mCerulean fluorescence; right: Venus fluorescence. **g** The reciprocal of f, in which USA300 expresses mCerulean fluorescence and G1905 or G1905 pCT545::pCURE express Venus fluorescence. **f**, **e**, **g** *n* = 3 biological replicates, mean ± standard error. Images of indicator plates are representative of three independent experiments. Source data are provided as a Source Data file.

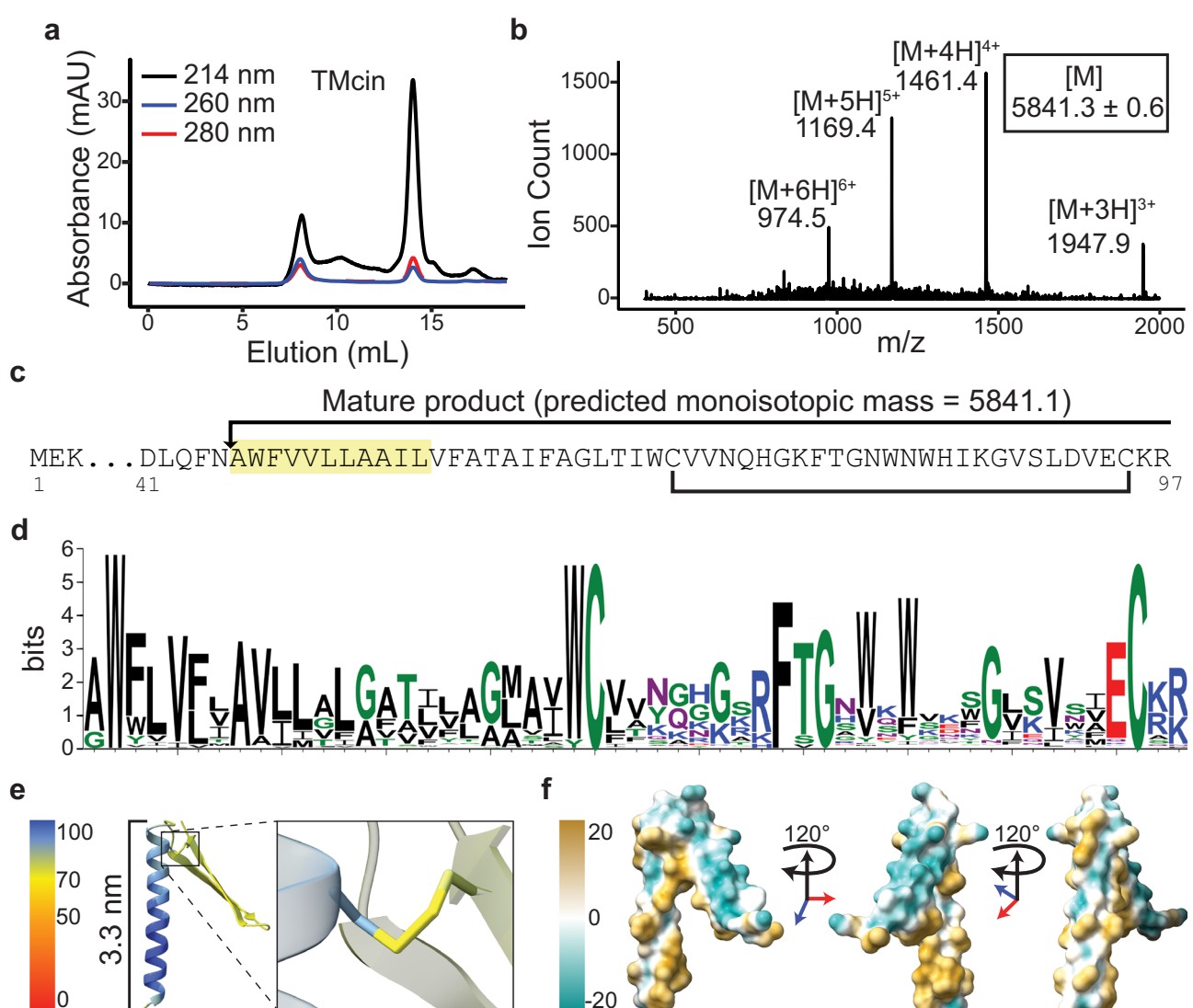

**Fig. 2 | Identification of the TMcin AMP. a** Size-exclusion chromatography of the purified antimicrobial substance on a Superdex 75 column. **b** Quadrupole mass spectrometry analysis of the purified antimicrobial substance. [M] is the deconvoluted monoisotopic molecular weight (mean ± standard deviation). **c** N-terminal sequencing result (highlighted), which when combined with the molecular weight identifies the antimicrobial substance as a modified peptide that matches a pre-peptide encoding gene on the BGC (red arrow in Fig. 1b). Arrow denotes the cleavage site of the pre-peptide and the bracket indicates a disulfide bond between two conserved cysteine residues. **d** Sequence logo of the mature peptide derived from multiple sequence alignment. **e** AlphaFold2 predicted structure colored by the pLDDT score of the TMcin mature peptide showing a N-terminal TMH and a C-terminal two-stranded β-sheet loop, each end of which is pinned by a disulfide bond (right, magnified). **f** The molecular lipophilicity potential[79] of TMcin-G1905 showing an amphipathic fold (images rotated 120° around the vertical axis). Source data are provided as a Source Data file.

activity (Fig. 1c). However, we rescued antimicrobial activity by complementing G1905 pCT545::pCURE with the pCT545 putative BGC subcloned into a heterologous plasmid backbone. Moreover, the G1905 pCT545::pCURE strain was sensitized to killing by G1905, reflecting that immunity mechanisms are commonly encoded within BGCs[31] (Fig. 1d). Finally, G1905 competed better than G1905 pCT545::pCURE in co-culture against the MRSA strain USA300, which commonly causes infections throughout North America but does not encode for a similar BGC[32] (Fig. 1e–g). Taken together, these data provided genetic evidence that the pCT545 accessory genes contain a novel BGC that results in the biosynthesis of an antimicrobial molecule.

## Antimicrobial identification

We hypothesized that the unknown antimicrobial was a ribosomally produced peptide because the BGC lacked genes encoding enzymes typical of non-ribosomal peptide synthetases or polyketide synthases.

When we attempted to purify the antimicrobial substance using reverse-phase (RP) chromatography, we initially failed, possibly due to the high background of staphylococcal phenol-soluble modulin (PSM) peptides which elute at higher concentrations of organic solvent[33]. We eliminated this background by deleting the *agr* quorum-sensing locus in G1905, which is absolutely required for PSM expression (Supplementary Fig. 2a)[34]. Using the G1905 Δ*agr* mutant, we obtained a pure product with bactericidal activity using a two-step reverse-phase and gel-filtration chromatography purification strategy (Fig. 2a, Supplementary Fig. 2b). Mass spectrometry and N-terminal sequencing identified a 5.8-kDa peptide that matched a proteolytically processed product encoded by a 297-bp structural gene, which we named *tmcA*, on the pCT545 BGC (Fig. 2b, c). To confirm the identity of the antimicrobial, we synthesized the C-terminal segment, raised rabbit polyclonal sera, and used the sera to detect the 5.8-kDa peptide in purified preparations as well as in G1905 culture supernatants by western blot (Supplementary Fig. 2c, d). Notably, the TmcA gene

product was not included in the recently published AMPSphere collection that leveraged Macrel, a machine-learning pipeline, to predict nearly one million AMPs[14]. Even accounting for site-specific proteolysis, the Macrel AMP prediction server failed to classify the 5.8 kD TMcin peptide as a likely AMP (Supplementary Table 2)[35].

Remarkably, 65% of the mature peptide residues are hydrophobic and concentrated within the N-terminal half. This N-terminal hydrophobicity is conserved across homologs found throughout Gram-positive species (Fig. 2d; see further below on details regarding homologous systems). Several sequence-based algorithms, including AlphaFold2 (AF2)[36], predicted an α-helical TMH, flanked by two highly conserved tryptophan residues (Fig. 2d, Supplementary Fig. 3a–c)[37,38]. Nonetheless, the peptide is secreted and diffusible (Fig. 1a and Supplementary Fig. 2c). Given the TMH hallmark, along with the following bioinformatic analysis and mechanistic description, we refer to this family as the TMcin (transmembrane containing bacteriocin) family and to the specific homolog isolated from *S. aureus* G1905 as staphylococcal TMcin-G1905 or briefly, TMcin-G1905[39].

AF2 predicted that the 28 residues that follow the TMH adopt an amphipathic, two-stranded, antiparallel β-sheet in which each end is pinned near the TMH by a disulfide bond between conserved cysteine residues (Fig. 2d, e). The presence of a disulfide bond is supported by two observations: First, the putative BGC contains a gene with homology to genes encoding enzymes of the DsbA family, which are commonly responsible for ascertaining the formation of structurally correct intrachain disulfide bonds[40] (Figs. 1b, 2e). Second, we detected a gel-migration shift, indicative of a conformational change, after exposing TMcin-G1905 to reducing reagents (Supplementary Fig. 3d). Finally, the amphipathic β-sheet is predicted to fold over one face of the TMH, exposing the hydrophilic surface to the environment (Fig. 2f and Supplementary Fig. 3e), likely imparting an overall amphipathic property to TMcin-G1905. This feature helps to explain the - at least limited - aqueous solubility of TMcin-G1905, despite the presence of a considerable portion of hydrophobic amino acids in its sequence.

Using PSI-BLAST, we found TMcin homologs in the phyla Bacillota and Actinomycetota (Gram-positive bacteria) that phylogenetically clustered mainly by bacterial genus, although the deeper nodes in the tree were not well supported (Fig. 3a). Of note, TmcA does not bear any homology to known bacteriocins in the mature or leader peptide part. We also identified an operon, with three different gene arrangements, for all homologs as candidate BGCs. These putative BGCs corresponded to the largest operon from the TMcin-G1905 putative BGC and included a gene with homology to the structural gene, *tmcA*, a gene with homology to a metalloprotease, *tmcP*, and two genes with homology to ABC transporter subunits, *tmcBC* (Fig. 3b). Inferred by homology, these functions are consistent with ribosomal synthesis, proteolytic maturation, and export. In addition, the candidate BGCs that most frequently associated with the *Staphylococcus* genus, including the TMcin-G1905 BGC on pCT545, additionally contained a monocistronic operon encoding a DsbA family member and a bicistronic putative regulatory operon that includes a gene encoding a helix-turn-helix transcriptional regulator. Although the *S. aureus* G1905 putative BGC has been mobilized on a plasmid, several homologous systems were found on completed chromosomes (e.g., NZ_CP025935.1 and NC_006270.3). Finally, taking together the TMcin-G1905 processing events and identification of putative BGC-encoded post-translational modification enzymes, TMcin is a new RiPP family[41].

## Solubilization by amphipathic peptides

Interestingly, we noted that the G1905 Δ*agr* mutant produced a smaller zone of clearance than WT G1905 (Fig. 4a). Although less TMcin was present in G1905 Δ*agr* culture supernatants than G1905 WT, we recovered similar amounts from cell pellets using 8 M urea (Fig. 4b). We hypothesized that PSMs further increase TMcin solubility because PSMs, absent in the Δ*agr* mutant, are powerful surfactants capable of solubilizing lipoproteins and releasing membrane vesicles[42–44]. Consistent with this, purified TMcin-G1905 exhibited synergistic activity with synthetic PSMs (Fig. 4c). Notably, the PSM δ-toxin, which displayed the greatest synergy, had to be pre-mixed with TMcin-G1905 in the same well to yield a zone of clearance, while even very close separate application in different wells did not produce a zone of synergistic activity between the wells (Fig. 4d), indicating that the two peptides do not work synergistically on the target but one aids solubilization of the other. Moreover, we analyzed culture filtrates by size-exclusion chromatography and found that the TMcin population secreted by G1905 Δ*agr* shifted to higher molecular weight complexes in comparison to G1905 WT (Fig. 4e). Finally, using killing assays in well-mixed aqueous solutions, δ-toxin was not necessary for killing and only displayed synergy with TMcin-G1905 when present in excess (Fig. 5a). Taken together, these data support the notion that surfactants naturally secreted by the TMcin producer G1905 boost TMcin solubility and aid in its release from the cell surface.

## Mechanism of action

TMcin-G1905 exhibited broad-spectrum activity against Gram-positive bacteria, including drug-resistant MRSA and vancomycin-resistant *Enterococcus faecalis* (Fig. 5b). Importantly, no target strain was a TMcin producer or encoded a TMcin BGC. In contrast, the Gram-negatives *Escherichia coli* and *Pseudomonas aeruginosa* were resistant, suggesting that the outer membrane that is characteristically present in Gram-negative bacteria prevented TMcin from reaching the site of antimicrobial action.

Given the presence of a TMH, we suspected that TMcins exert antimicrobial activity by perturbing cell membranes. Although TMcin-G1905 did not lead to the dissolution of the cell membrane, we detected it in membrane preparations of treated MRSA cells (Fig. 5c), of which electron and confocal fluorescent microscopy revealed a shrunken cell phenotype with an apparent loss of turgor pressure (Fig. 5d). Moreover, TMcin-G1905-treated cells sustained a loss of membrane potential and released ATP (Fig. 5e, f). These data suggest the formation of membrane pores and are reminiscent of a recently proposed biophysical model in which persistent nanometer-scale membrane defects lead to a loss of cellular solutes, membrane polarization, turgor pressure, and cell volume[45].

To test for pore formation directly, we first confirmed that TMcin-G1905 acts on the phospholipid membranes of liposomes, in which it mediated the leakage of the membrane-impermeant carboxyfluorescein dye out of loaded liposomes (Fig. 6a). We then measured TMcin-G1905-induced time-dependent current using voltage-clamped planar lipid bilayers. Adding purified TMcin-G1905 resulted in increasing conductance over the course of several minutes that was comprised of discrete stepwise events with a large median conductance of 10.8 nS in 150 mM KCl (Fig. 6b, c), pointing to the formation of ion-conductive pores. Importantly, these pores showed linear (Ohmic) current-voltage dependencies and no measurable ion selectivity (Fig. 6d). As a comparison, pores formed by gramicidin A and the fibupeptide lugdunin, both 0.4 nm in diameter, yielded only 30 pS even though higher salt concentrations of 0.5 M KCl were used[18,46], underscoring the exceptional stability of TMcin pores that resembles that of larger pore-forming toxins.

The large conductance of the TMcin-G1905-induced pores in a sub-molar salt solution and the virtual lack of ion selectivity indicated that the pores are large[47] and, therefore, oligomeric. Moreover, the persistent increase of total conductance suggested a discrete pore architecture that is held together through intermolecular interactions and was reminiscent more of well-structured pore-forming proteins[48–50] than of membrane-permeabilizing peptides[51–55]. Without

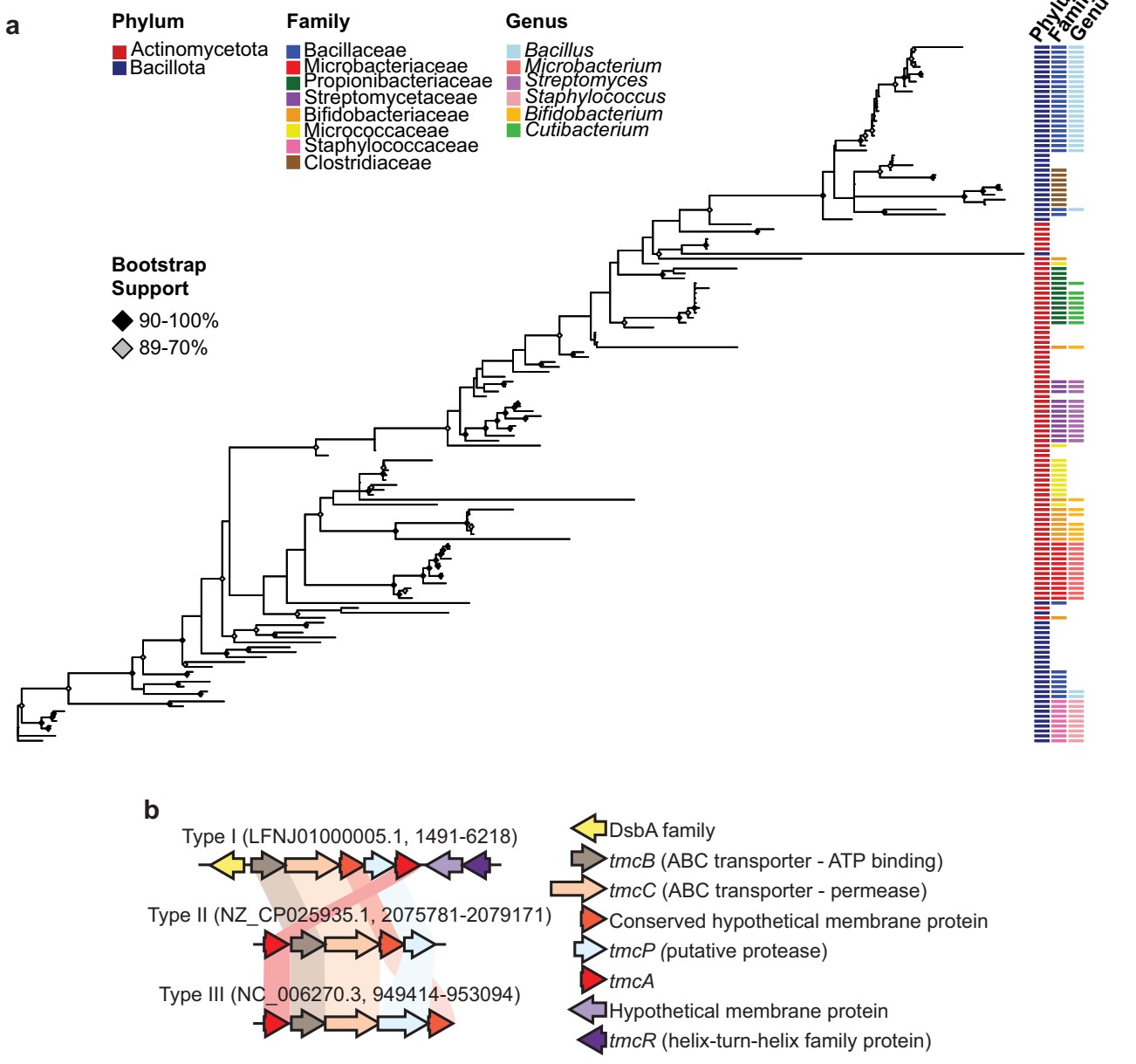

**Fig. 3 | The TMcin family is broadly distributed throughout Gram-positive bacteria. a** Unrooted phylogenetic tree of TmcA (pre-TMcin protein). Node support is indicated by shaded diamonds. The most abundant bacterial phylum, family, and genus taxonomic groups are marked with colored boxes to the right of TmcA taxa. **b** Representative operon structures of TMcin BGCs (example accession number, BGC locus). Source data are provided as a Source Data file.

any a-priori insight into the TMcin pore stoichiometry, we adopted an agnostic approach and queried AF2 using all n-mers between 2 and 30 to predict pore structures. Remarkably, we acquired plausible ring-like pore structures of between 10 to 25 (except 24) TMcin-G1905 protomers (Supplementary Fig. 4a), of which the pLDDT scores were in the confident range (70 < pLDDT < 90) for n-mers between 15 and 23. However, the model confidence scores were generally low (<0.4). We therefore used the highest scoring 21-mer pore model as a template for AF2 and achieved similar ring-like models up to 30-mers with improved metrics (Fig. 6e and Supplementary Fig. 4b). In all models with $n \geq 14$, the TMHs formed a hydrophobic belt that surrounded a β-barrel hydrophilic interior (Fig. 6f). The first β-strand of one protomer hydrogen bonded and formed a salt bridge with the second β-strand of the neighboring protomer (Fig. 6g).

To further assess the AF2 predicted structures, we ran molecular dynamics (MD) simulations of the TMcin-G1905 monomer in solution and the predicted 21-mer oligomeric pore in a lipid bilayer environment that models the target Gram-positive cellular membrane. The AF2 model of the TMcin monomer retained its general fold during the 1-µs MD simulations (Fig. 6h; Supplementary Fig. 5a,b; Supplementary Video 1). The monomeric units of the TMcin pore complex embedded in a 7:3 DMPG:cardiolipin bilayer maintained their initial structures to a greater extent than the monomer in solution (Fig. 6i; Supplementary Fig. 5; Supplementary Video 2), with better preservation of secondary structure, and lower fluctuations in atom position (Fig. 6j) and less deviation from the initial structure (Fig. 6k). The pore complex varied in circularity between replicas, but remained intact over the course of the 0.5 µs simulations (Fig. 6l, Supplementary Video 2). Thus, both the TMcin monomer and the oligomeric pore retained the fold predicted by AF2 throughout the µs-long MD simulations in solution and the 0.5 µs-long simulations in a lipid bilayer. A subset of residues within the monomer structure exhibited substantial fluctuation; however, this

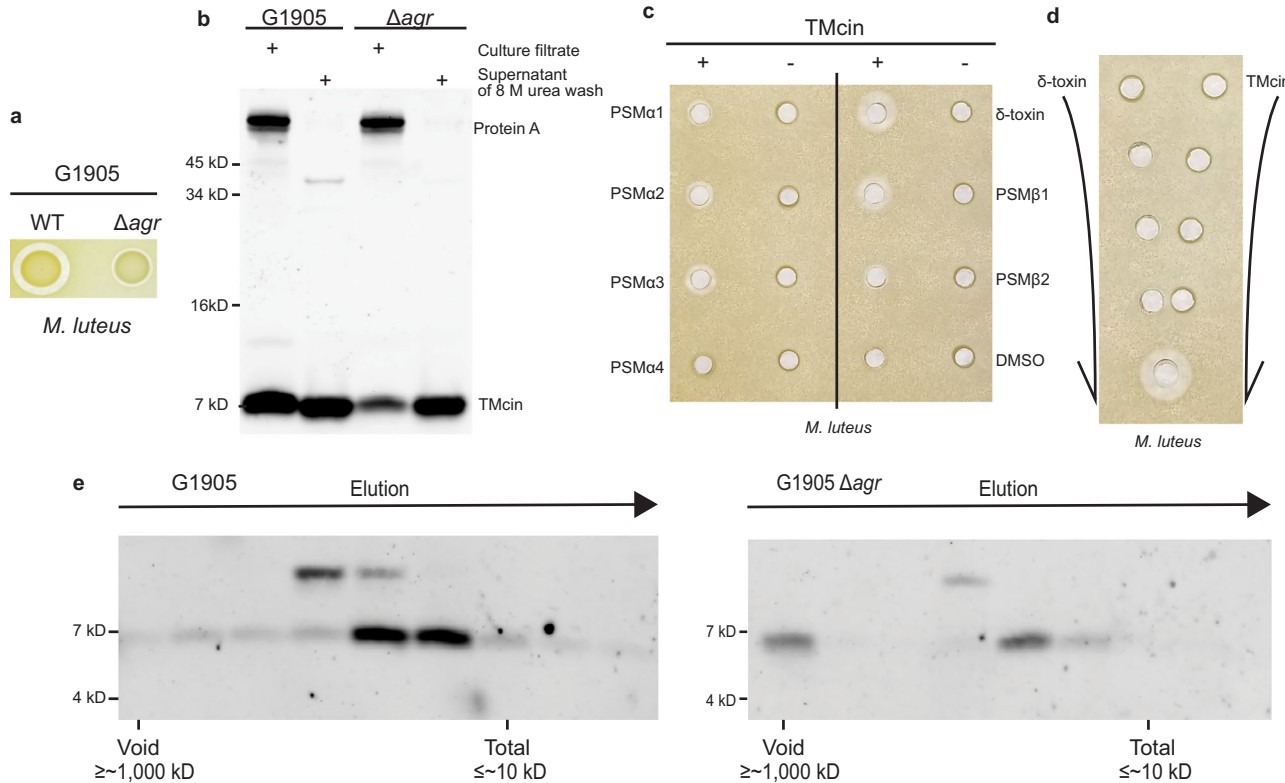

**Fig. 4 | *S. aureus* surfactant peptides boost solubilization of TMcin-G1905.**
**a** Antimicrobial indicator assays of G1905 and G1905 Δ*agr*. **b** G1905 and G1905 Δ*agr* secreted TMcin-G1905 in culture supernatants or after washing cell pellets with 8 M urea. **c** Antimicrobial indicator assay demonstrating synergistic activity between TMcin-G1905 and individual *S. aureus* PSMs. Wells containing TMcin-G1905 were mixed with (+) or without (−) the indicated PSM. **d** Antimicrobial indicator plate demonstrating a pre-mixing requirement for TMcin-G1905 synergism with δ-toxin,

suggesting that PSMs aid in solubilizing TMcin-G1905 to enable it to diffuse through the agar matrix. **e** Superdex 200 size-exclusion chromatography of G1905 and G1905 Δ*agr* secretions into culture filtrates. Fractions were analyzed by western blot using αTMcin-G1905 rabbit antisera. Images are representative of two (western blots) or three (indicator plates) independent experiments. Source data are provided as a Source Data file.

was expected given the small size of the peptide. Notably, all residues within the 21-mer pore structure exhibited little fluctuation, which is likely due to the non-polar lipid environment that promotes secondary structure and the extensive interactions between β-strands of the protomers that form the β-barrel.

We also analyzed TMcin-G1905 in solution using circular dichroism spectroscopy (CD). The CD results revealed both α-helical and β-sheet secondary structures, which supports the AF2 models and the MD simulations (Fig. 6m and Supplementary Table 3). Interestingly, the CD spectrum also demonstrated a substantial fraction of disordered residues, which is consistent with the monomer MD simulation that showed fluctuations of residues within the β strands and the N-terminus of the α-helix.

The large pore structures predicted by AF2 and MD simulations would form large water-filled pores ranging up to 5 nm in radius. To determine whether these structures exist on the membranes of treated cells, we tested the permeability of large polymers with average Stokes-Einstein radii ($\bar{r}_{ES}$) of up to 4.7 nm. Indeed, MRSA cells treated with TMcin-G1905 became fluorescent with both the linear polymer FITC-dextran 40,000 ($\bar{r}_{ES}$ ~4.7 nm) and the globular polymer FITC-polysucrose 40,000 ($\bar{r}_{ES}$ ~3.9 nm) (Fig. 7a, b). Importantly, we confirmed cellular uptake of the FITC-polysucrose polymer by confocal imaging in which untreated cells excluded the polymer whereas TMcin-G1905 treated cells internalized it (Fig. 7c).

## Discussion
Herein, we report the discovery of an unusual family of bacteriocin AMPs, which we termed TMcin. The most striking feature of TMcins is

that they form pores in membranes that are remarkably stable and large. While similar pores have been observed among the much larger pore-forming proteins, stable AMP pores have been limited to small, sub-nanometer channels[18,56]. In contrast, AF2 predicted plausible pore architectures that consisted of 10 to 30 TMcin-G1905 copies in which the TMHs encircled an inner β-barrel pore that formed membrane-spanning aqueous channels of up to 10 nm in diameter. While AF2 has proven remarkably accurate, especially in predicting secondary structure of mixed secondary structure peptides and for α-helical transmembrane segments[57], deviations from experimental structures have been noted[58–61]. We therefore also ran MD simulations and performed CD spectrometry, which supported the AF2 predictions. In addition, flow cytometry and confocal microscopy confirmed that large macromolecular fluorescent polymers readily permeated cells treated with TMcin-G1905. Thus, our in-silico simulations, biophysical, and microbiological assays support the existence of these pores. Taken together, our results provide previously unavailable evidence for the existence of a family of nanoscale pore-forming AMPs. Examples of such AMPs have been described but their existence has not been supported with explicit structural models[19,62].

The eponymous TMH is a prominent feature of the TMcin family. Although TMHs usually anchor proteins in cell membranes, antimicrobially active TMcin-G1905 diffused away from producing cells. Accordingly, a conserved ABC transporter gene within the TMcin BGC suggests that TMcin is exported through the cytoplasmic membrane. Upon secretion, AF2 structural models predict TMcin maintains solubility by assuming an amphipathic fold in which the C-terminal β-sheet drapes over one face of the TMH. However, during our purification

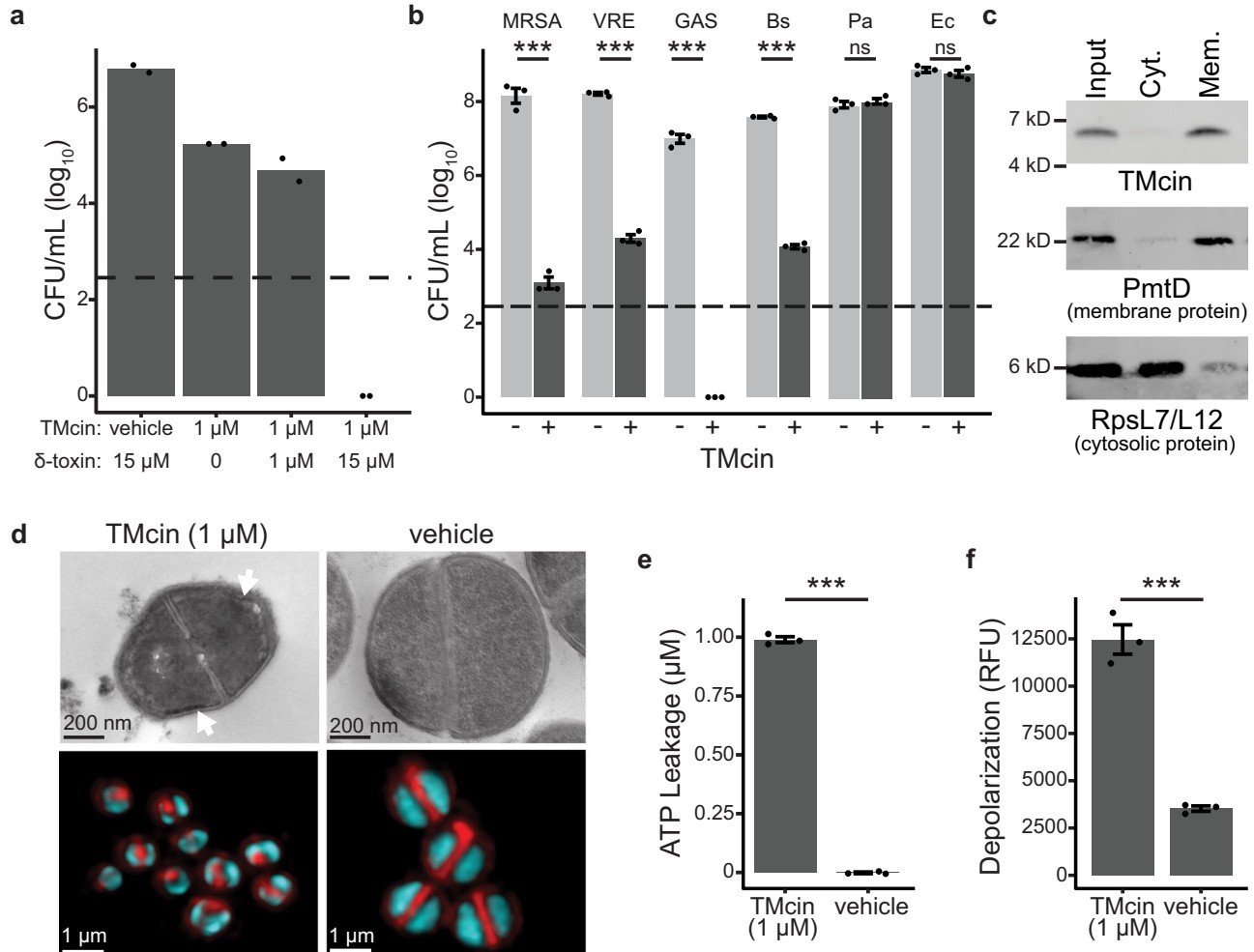

**Fig. 5 | TMcin targets cell membranes and releases cellular contents.**
**a** Synergistic activity of TMcin-G1905 with the staphylococcal surfactant peptide δ-toxin. Excess, but not equimolar δ-toxin led to synergistic killing by purified TMcin-G1905 of MRSA USA300 in 1 h. ($n = 2$; dashed line and data points below represent limit of detection and no colonies observed, respectively). **b** Gram-positive bacteria exhibited sensitivity to killing with $1\,\mu M$ purified TMcin-G1905 (+) compared with vehicle (−). ($n = 3$ biological replicates, mean ± standard error; ***, $p < 0.001$; ns, $p > 0.05$ for pairwise comparisons using Tukey's HSD test, family-wise α = 0.05). **c** Subcellular fractionation of USA300 cells after treatment with purified TMcin-

G1905 showing localization to the cell membrane. **d** Thin-section transmission electron microscopy (top) and confocal fluorescence microscopy (below; FM 4-64 membrane stain, red; SYTO-9 DNA stain, cyan) of USA300 cells after treatment with purified TMcin-G1905. White arrow in EM images indicates membrane ruffling that may be caused by loss or turgor pressure resulting in a loss of membrane tension. **e** ATP leakage from and **f** membrane depolarization of USA300 cells after treatment with purified TMcin-G1905 ($n = 3$ biological replicates, mean ± standard error; ***, $p < 0.001$ by one-way ANOVA). Source data are provided as a Source Data file.

procedures, we noticed that its aqueous solubility was limited. Interestingly, in staphylococci, TMcin solubility was boosted by co-produced surfactant peptides[63] to overcome that limitation. It is tempting to speculate and remains to be explored whether the bacteria in which TMcin BCGs are frequently found produce similar surfactants or whether they live in an environment containing surfactant-like molecules that aid in solubilizing TMcins. It will also be interesting to decipher why only some rare strains of staphylococci produce TMcins and whether TMcin production gives an in vivo competitive advantage in the specific infectious environment from which they have been isolated.

TMcin eluded prior identification because both TMcin and the TMcin BGC are sufficiently distinct from other antimicrobials and their BGCs[41,64] to conceal TMcin from identification by genome-mining approaches. Upon our identification, we detected the TMcin family throughout two major bacterial phyla. Thus, despite impressive advances in predicting antimicrobials[9,12,14,30], our discovery hints that additional antimicrobial classes with unforeseen features and mechanisms exist and will require phenotype-initiated investigations to be uncovered.

Taken together, our discovery of TMcin and our mechanistic insights represent important extensions to our understanding of AMP structure and function, inasmuch as they blur the line between dynamic membrane-permeabilizing peptides and stable pore-forming proteins. An array of approaches is needed to address the burden of antimicrobial resistance that is expected to worsen in the coming decades[65]. The TMcin family not only offers new AMP candidates to treat infections but also provides a novel mechanistic framework to build upon and harness the potential of nanometer-scale pore-forming AMPs.

## Methods
### Ethical statement
Animal work was approved by the Institutional Animal Care and Use Committee of the NIAID (study protocol LB1E). Animal work was conducted by certified staff in a facility accredited by the Association for Assessment and Accreditation of Laboratory Animal Care (AAALAC). All of the animal work adhered to the institution's guidelines for animal use and followed the guidelines and

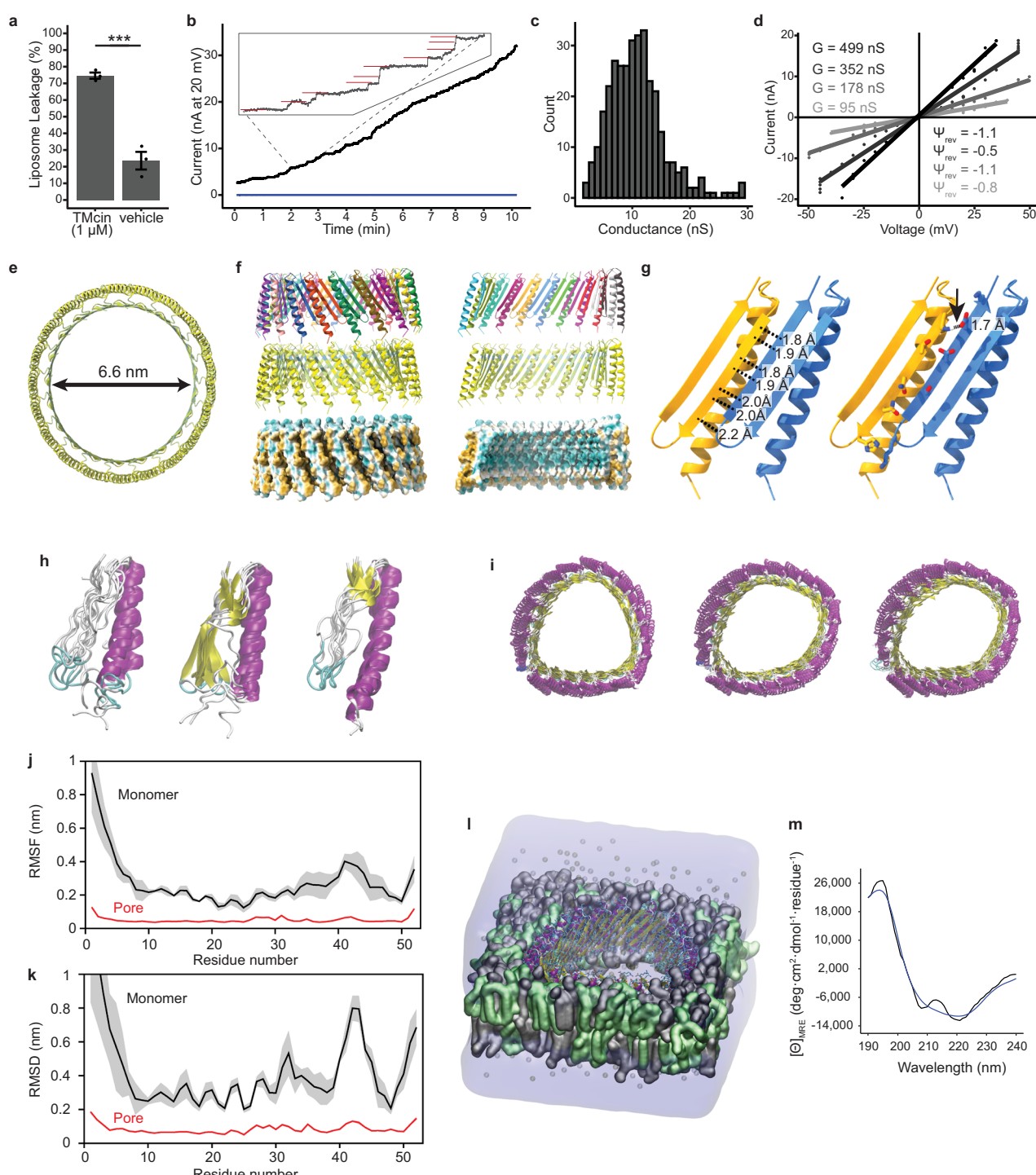

basic principles in the US Public Health Service Policy on Humane Care and Use of Laboratory Animals, and the Guide for the Care and Use of Laboratory Animals.

### Bacterial isolates and growth conditions

Bacterial strains are listed in Supplementary Table 4. All cells were cultured using tryptic soy broth (TSB) at 37 °C on a shaking platform unless otherwise indicated. *Streptococcus pyogenes* was cultured statically at 37 °C and 5% $CO_2$ in Todd Hewitt broth supplemented with 0.2% yeast extract (THY). Working concentrations of kanamycin, erythromycin, and chloramphenicol were used at 100, 10, and 10 µg/mL, respectively.

### Molecular cloning and allelic exchange

Q5 polymerase (NEB) was used for all PCR amplifications. To cure G1905 of pCT545, the pCT545 toxin/antitoxin and replication genes were amplified using pCT545_NotI_F and pCT545_KpnI_R oligonucleotides (Supplementary Table 5) and pCT545 as template. The *aph* gene was amplified from pKX15[66] using aph_NotI_F and aph_R oligonucleotides. PCR products were digested with NotI-HF and KpnI-HF and ligated using T4 DNA ligase. The ligation product was electroporated into G1905, and transformed cells were selected on tryptic soy agar (TSA) supplemented with kanamycin. Cells were passaged once by a 1000-fold dilution into fresh TSB supplemented with 100 µg/mL kanamycin at 37 °C. A single colony was isolated by plating serial

**Fig. 6 | TMcin oligomerizes to form β-barrel pores. a** TMcin-G1905 mediated leakage of carboxyfluorescein-loaded liposomes (7:3 DMPG:cardiolipin molar ratio) relative to Triton X-100 (100%). ($n$ = 3 biological replicates, mean ± standard error; ***, $p < 0.001$). **b** A representative current trace through planar lipid bilayer formed from DPhPC in 150 mM KCl, pH 7.4 after addition of 11.5 nM TMcin-G1905 to the cis compartment. Blue line indicates current after equal-volume addition of the mock-purified control to the cis compartment. Inset (top-left) shows individual stepwise current increase events marked by red horizontal lines. **c** Histogram of stepwise conductance events ($n$ = 332). **d** Current-voltage (I/V) curves obtained for 4 different conductance levels (G) induced by TMcin-G1905 in planar membranes made in 150 mM (cis) / 730 mM (trans) KCl gradient. The reversal potential ($\Psi_{rev}$) was calculated as the intercept of the linear regression fits for each conductance level. **e** Top-view of a 21-mer TMcin pore model colored by pLDDT (for scale see Fig. 1i). **f** Lateral view and cross-section (right, 11 protomers visible) of the 21-mer pore model colored by chain (top), pLDDT (middle), and molecular lipophilicity

potential (bottom). **g** Ribbon model showing backbone hydrogen bonding (left) and a salt bridge (right, arrow, Lys 32 and Glu 49) between β-strands of neighboring protomers. **h**–**l** MD simulation analysis. **h, i** Overlays of representative structures spaced uniformly throughout the trajectories of the monomeric TMcin peptide in solution (**h**) and pore complex in a bilayer (**i**), colored by secondary structure. **j** Cα RMSF values for the monomer (black) and monomeric units of the pore complex (red). **k** Cα RMSD values for the monomer (black) and monomeric units of the pore complex (red). **j, k** $n$ = 3 independent simulations, mean (line) ± standard deviation (shade). **l** Renderings of the pore complex system after 0.5 μs (replica 1), showing the TMcin pore, lipids (DMPG and cardiolipin in grey and green, respectively), water (transparent blue), and ions (K and Cl in silver and gold, respectively). **m** Circular dichroism analysis. Far-UV mean residue ellipticity of 2 μM TMcin-G1905 in phosphate buffer and 5% acetonitrile (black) fitted with the closet match of a linear combination of reference proteins to estimate secondary structural components (blue). Source data are provided as a Source Data file.

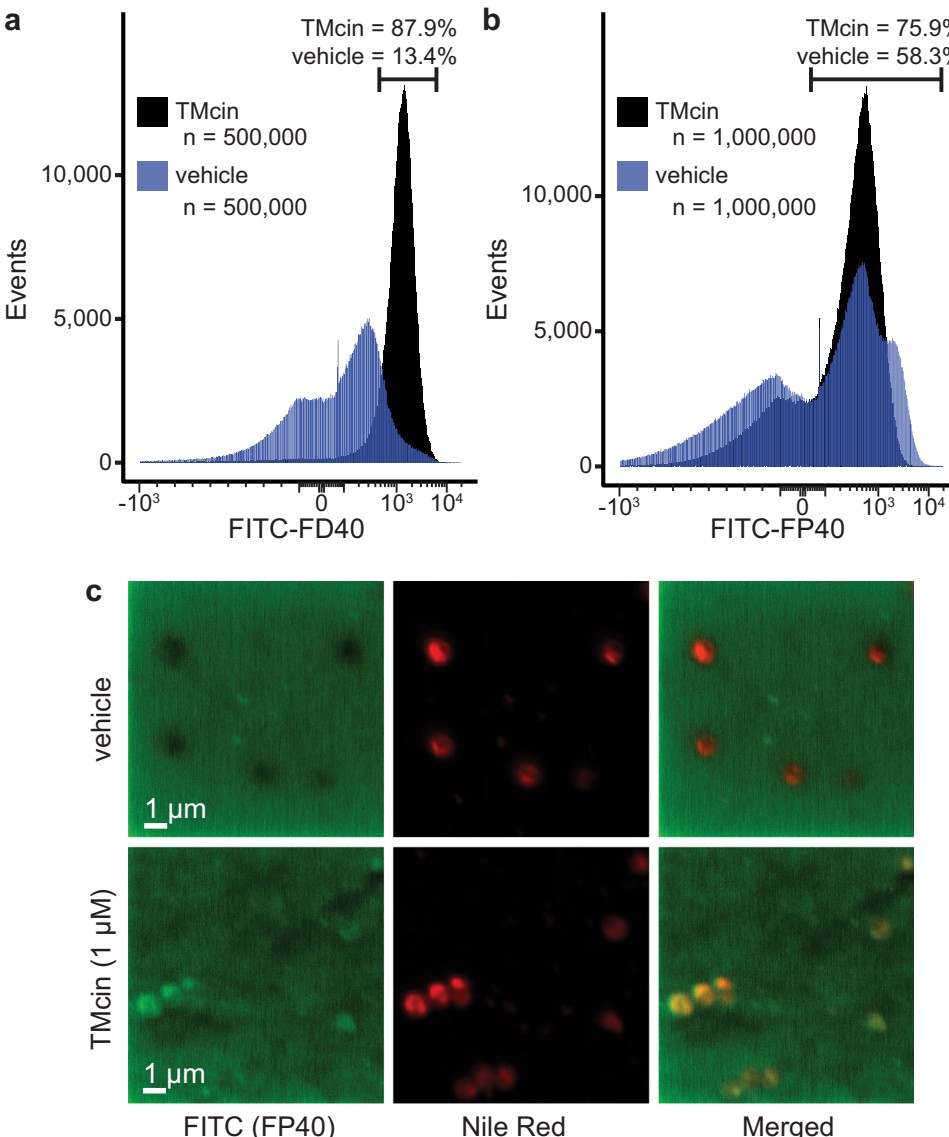

**Fig. 7 | TMcin permeabilizes MRSA cells to macromolecules. a** Flow cytometry histogram of TMcin- or vehicle-treated USA300 cells after incubation with FITC-dextran 40,000 (FITC-FD40) or **b** FITC-polysucrose 40,000 (FITC-FP40).

**c** Confocal images of TMcin- or vehicle-treated USA300 cells. After incubation with FITC-FP40, cells were stained with Nile Red to stain membranes. Images are representative of three independent experiments.

dilutions on TSA. Loss of pCT545 was confirmed by PCR genotyping *tmcA* using tmcA_us_F and tmcA_ds_R oligonucleotides.

The complementation plasmid pRB + BGC was constructed as follows. A putative ST88 type 1 restriction site[67] was removed from

pRB573[68] by inverse PCR using Remove_ST88_RS_R and Remove_ST88_RS_L and circularization with T4 PNK and T4 DNA ligase (NEB). The pCT545 BGC was amplified from a G1905 genomic DNA preparation using pCT545_BGC_EcoRI_F and pCT545_BGC_BamHI_R. The PCR

product and pRB573 were digested with EcoRI and BamHI and ligated using T4 DNA ligase. The ligated reaction was electroporated into G1905 pCT545::pCURE cells. The correct clone was confirmed by PCR using pRB_F and pRB_R and by Sanger sequencing.

The *agr* locus was deleted from G1905 using pIMAY allelic exchange[69]. G1905 is naturally chloramphenicol-resistant; therefore, an erythromycin-resistant pIMAYermC was constructed. pIMAY was amplified using pIMAY_ds_cat_L and pIMAY_Phelp_R. The *ermC* gene was amplified from pJL_sar_GFP[70] using ermC_F and ermC_R, and the PCR product was treated with T4 polynucleotide kinase (PNK, NEB). PCR products were ligated (T4 DNA ligase, NEB) and transformed into *Escherichia coli* DC10B cells (erythromycin working concentration = 250 µg/mL). Isolated plasmid from a single colony was confirmed by Sanger sequencing. The *agr* locus and 1 kb upstream and downstream were amplified from G1905 genomic DNA and inserted into the EcoRV site of pIMAYermC. Subsequently, *agr* was deleted by inverse PCR using G1905_Δagr_L and G1905_Δagr_R and circularization using T4 PNK and T4 DNA ligase. The pIMAYermC Δ*agr* plasmid was electroporated into G1905 and G1905 pCT545::pCURE. Following allelic exchange, G1905 Δ*agr* clones were confirmed by PCR using G1905_agr_1k_R and G1905_agr_locus_F, the latter of which does not anneal to pIMAYermC Δ*agr*.

## TMcin-G1905 purification

**Sample preparation.** G1905 Δ*agr* cells were cultured in 500 mL of TSB starting at an $OD_{600}$ of 0.1 for 8 hours at 37 °C on a shaking platform. Cells were collected by centrifugation (JLA 10.500 rotor, Beckmann Coulter), and resuspended with 20 mL ultra-pure $H_2O$. The cells were pelleted again by centrifugation and resuspended with 20 mL 8 M urea. The cells were pelleted by centrifugation, and the supernatant was recovered and filtered through a 0.2 µm polyethersulfone (PES) filter.

**Reverse-phase purification.** Initially, a C18 Symmetry 300 Å column was connected to an Äkta pure 25 chromatography system (Cytiva, Unicorn version 7.5.0) and equilibrated with buffer A composed of ultra-pure $H_2O$, 0.1% trifluoroacetic acid (TFA). The 8 M urea cell-wash filtrate was injected onto the column, which was subsequently washed with two column volumes (CV) of buffer A. A 10 CV gradient of 0–100% buffer B, consisting of isopropanol, 0.1% TFA, was applied and eluted fractions were collected. Fractions were frozen at −80 °C and lyophilized (−84 °C freeze dryer, Labconco).

Because residual TMcin-G1905 remained bound to the C18 column and continued to elute after multiple gradients, we now routinely use C18-solid phase extraction (SPE) columns (Waters #WAT036905) attached to a vacuum manifold (Waters #WAT200609). The SPE column was conditioned with 10 mL buffer B and equilibrated with 10 mL 10% buffer B, after which 1.8 mL of the 8 M urea cell-wash filtrate was applied. The column was washed with 10 mL each of 10% and 40% buffer B. TMcin-G1905 was eluted and collected using 6 mL 65% buffer B. The column was washed with 10 mL 100% buffer B. Equilibration, sample application, washes, and elution were repeated until all sample was processed. Eluted fractions were pooled, frozen, and lyophilized.

**Size exclusion chromatography.** Lyophilized samples were resuspended with 50% acetonitrile (ACN), 0.1% TFA, and 100 µL was injected onto a Superdex 75 column (Cytiva) that was equilibrated with 50% ACN, 0.1% TFA. TMcin-G1905 was eluted and collected with running buffer composed of 50% ACN, 0.1% TFA, and then frozen and lyophilized. Dried samples were resuspended to approximately 20 µM TMcin-G1905 in 50% ACN, 0.1% TFA. Aliquots were prepared and stored at −80 °C.

The concentration of purified TMcin-G1905 stock solutions was determined to be 20 µM by comparing the peak area of UV absorbance at 214 nm of 0.1 mg/mL synthetic PSMα3 vs TMcin-G1905 after

injecting each using an overfilled 50 µL capillary loop onto a Superdex 75 column equilibrated with 50% ACN, 0.1% TFA and adjusting for extinction coefficients predicted by sequence[71,72]: 288,870 $M^{-1*}cm^{-1}$, TMcin-G1905; 101,458 $M^{-1*}cm^{-1}$, PSMα3. Subsequent TMcin-G1905 preparations were resuspended to similar concentrations based on the peak area of the size-exclusion step and confirmed with previous TMcin-G1905 preparations by silver stains and western blots using αTMcin-G1905 rabbit antisera.

## αTMcin-G1905 rabbit antisera

The C-terminus of TMcin-G1905 (C25 to R52) was synthesized (Peptide 2.0) and resuspended in DMSO to 10 mg/mL. Two- to three-month-old New Zealand rabbits (Charles River Laboratories) were injected subcutaneously with 300 µg peptide each for a prime (with complete Freund's adjuvant), and four boosts (with incomplete Freund's adjuvant). Each injection was two weeks apart, and terminal bleeds were collected one week after the final boost. Immunogenicity was confirmed by ELISA using the synthetic C-terminal peptide. The animal protocol was approved by the Division of Intramural Research Animal Care and Use Committee of the National Institute of Allergy and Infectious Diseases (study protocol LB2E).

## N-terminal sequencing and mass spectrometry

A lyophilized fraction of TMcin-G1905 after reverse-phase purification was submitted for N-terminal sequencing by Edman degradation (NIAID Research Technology Branch). Purified TMcin-G1905 was injected directly into an electrospray source connected to a quadrupole mass spectrometer (Agilent 6120B) set to scan a m/z range of 400–2000 in the positive mode. The deconvoluted mass (M) for each m/z peak with charge (i) of +3 to +6 was calculated by $M_i = (m/z_i - 1.007)*i$, from which the mean (M = $\frac{\sum_{i=3}^{6} M_i}{4}$) and standard deviation (s = $\sqrt{\frac{\sum_{i=3}^{6}(M_i - M)^2}{3}}$) were derived.

## Cell-killing assays

Cells were grown to late-log phase, pelleted, and resuspended in assay buffer (10 mM $KPO_4$, pH 7.0, 5 mM MgSO4, 0.25 M sucrose). Cell density was adjusted to an $OD_{600}$ of 0.5. Purified TMcin-G1905 or vehicle (50% ACN, 0.1% TFA) was diluted 10-fold into assay buffer supplemented with 0.1 mg/mL δ-toxin. Equal volumes of cells were mixed with TMcin-G1905 or vehicle and incubated at 37 °C for 1 h. Treated cells were serially diluted 10-fold and spotted on tryptic soy agar (TSA) or THY-agar (*S. pyogenes* only) and incubated overnight to enumerate CFUs as a measure of viable cells.

For the δ-toxin titration, TMcin-G1905 (20 µM) or vehicle was diluted 10-fold in assay buffer supplemented with 0, 0.1 mg/mL (30 µM), or 0.067 mg/mL (2 µM) δ-toxin. *S. aureus* USA300 LAC cells were prepared as described and mixed in equal volumes with TMcin-G1905 or vehicle and incubated for 1 h before enumerating viable cells by serial dilution and spotting on TSA plates. Limit of detection was estimated at 285 CFU/mL to account for one ten-fold dilution and the volume spotted on TSA.

## Antimicrobial indicator plate assay

Indicator bacteria were mixed at $5 \times 10^5$ CFU/mL in molten TSA (7.5 g/L bacto-agar) cooled to 46 °C, and 15 mL each was poured into 100 mm sterile petri dishes and allowed to set and air dry. Test bacteria were cultured overnight at 37 °C and 20 µL spotted onto indicator plates, the spots were allowed to dry, and the plates were incubated for 24–48 h at 37 °C.

For purified TMcin-G1905 and synthetic PSMs, synthetic PSMs with N-formylation were custom synthesized (Peptide 2.0) and dissolved in DMSO to 10 mg/mL and diluted to 0.2 mg/mL in assay buffer. Wells in indicator plates were created by excising agar discs using a

3-mm biopsy punch. In each well 10 μL of 0.2 mg/mL PSMs or DMSO in assay buffer were mixed with 2 μL of 20 μM TMcin-G1905 in 50% ACN, 0.1% TFA. Plates were incubated at 37 °C for 24 h and analyzed for a zone of clearance of the indicator bacteria.

## Co-culture competition assays

G1905, G1905 pCT545::pCURE, and USA300 LAC were transformed with pJL-sarA-CER (Cerulean) or pJL-sarA-VEN (Venus)[70]. Cells were grown in TSB supplemented with erythromycin and adjusted to a cell density of $1 \times 10^5$ cells/mL. Pairwise combinations of cells were mixed in equal volume in a 96-well microtiter plate, cultured at 37 °C with constant shaking, and monitored for fluorescence in a Tecan Spark microplate reader (Cerulean, monochromators: $434 \pm 10$ nm, excitation and $485 \pm 20$ nm, emission; Venus, monochromators: $503 \pm 5$ nm, excitation and $535 \pm 20$ nm, emission).

## ATP leakage assay

Purified TMcin-G1905 supplemented with 0.1 mg/mL δ-toxin and *S. aureus* USA300 LAC cells were prepared, and the cells were treated for 15 minutes at 37 °C with 1 μM TMcin-G1905 or vehicle (see Cell-killing assays). Cells were removed by centrifugation, and ATP in the supernatant was detected using bioluminescence with recombinant firefly luciferase (ThermoFisher #A22066) according to the manufacturer's instructions. ATP was quantified against a standard curve using the supplied ATP. Luminescence was measured in a GloMax plate reader (Promega).

## Membrane depolarization assay

DiSC$_3$(5) (TCI America, #D4456) was prepared to 3 mM in DMSO and diluted to 80 μM in assay buffer. Purified TMcin-G1905 supplemented with 0.1 mg/mL δ-toxin and *S. aureus* USA300 LAC cells adjusted to an OD$_{600}$ = 0.05 were prepared (see Cell-killing assays). Cells were mixed with 4 μM DiSC$_3$(5) and monitored for fluorescence in a Tecan Spark microplate reader (monochromators: $620 \pm 15$ nm, excitation; $685 \pm 15$ nm, emission) until a stable baseline was achieved. TMcin-G1905 or vehicle was mixed 1:1 with DiSC$_3$(5)-loaded cells, and fluorescence was monitored for 30 minutes.

## Subcellular fractionation

Purified TMcin-G1905 supplemented with 0.1 mg/mL δ-toxin and *S. aureus* USA300 LAC cells adjusted to an OD$_{600}$ = 10 were prepared (see Cell-killing assays). Cells were mixed with TMcin-G1905 or vehicle at a 5:2 ratio ([TMcin-G1905]$_{final}$ = 0.6 μM) at 37 °C for 30 min. Cells were collected by centrifugation and washed twice in assay buffer to remove extracellular TMcin-G1905. Cells were lysed with acid-washed <100 μm glass beads (Sigma Aldrich, #G4649) using a FastPrep homogenizer (MP Biomedical) with 6 cycles of: 1) 6 m/s for 20 s, and 2) incubation on ice for 30 s. Beads were pelleted by centrifugation at $1500 \times g$ for 5 min and the supernatant was collected and further centrifuged at $30,000 \times g$ (TLA 120.1 rotor, Beckman Coulter) for 15 min. This supernatant was collected as the cytoplasmic fraction and the pellet was resuspended in assay buffer and centrifuged again at $30,000 \times g$ for 15 min. The subsequent pellet was resuspended SDS-PAGE sample buffer using a volume equivalent to the input (i.e. no concentration or dilution compared to cell lysis step). The fractions were analyzed by western blot for TMcin-G1905 (αTMcin-G1905 rabbit antisera), the ribosomal proteins RpsL 7/10 (Abcam #ab225681), and the membrane protein PmtD (2D1 hybridoma)[73].

## TMcin-G1905 size distribution

G1905 and G1905 Δ*agr* were grown in TSB at 37 °C for 7 h on a shaking platform, after which cells were pelleted centrifugation and the supernatant was filtered using a 0.2 μm PES filter. The culture filtrate was immediately injected using an overfilled 500 μL capillary loop onto a Superdex 200 increase column (Cytiva) equilibrated with 20 mM MES, pH 5.2, 150 mM NaCl running buffer. One-mL fractions were

eluted and collected at a flow rate of 0.75 mL/minute in running buffer. The fractions were precipitated with 15% trichloroacetic acid, incubated at 4 °C overnight. Precipitates were pelleted by centrifugation, washed with cold acetone, and resuspended with SDS sample buffer (30x concentration). Samples were analyzed by western blot using αTMcin-G1905 antisera after electrophoresis of samples with 16% SDS-PAGE in tricine running buffer and electrotransfer to 0.2 μm nitrocellulose membranes.

## Confocal microscopy

Purified TMcin-G1905 supplemented with 0.1 mg/mL δ-toxin and *S. aureus* USA300 LAC cells were prepared. Cells were treated for 30 min at 37 °C with 1 μM TMcin-G1905 or vehicle (see Cell-killing assays). Treated cells were stained with 7.5 μM FM 4-64 and 50 nM SYTO-9 at room temperature. Stained bacteria were added to an optical glass dish (Ibidi), and a 1 cm × 1 cm 1% agarose pad was placed on top of the cells.

Images were acquired on a Leica SP8 confocal microscope equipped with a 63X/1.4NA objective, 488n Argon laser, and tunable HyD detectors. Images were acquired with a voxel size of 24 nm x 24 nm x 200 nm. Sequential Z-sections of stained cells were deconvolved with Huygens Professional (version23.10.0-p5, SVI, Netherlands) and analyzed with IMARIS software (version 10.1, Andor Technology Inc., Concord, MA).

## Thin section transmission electron microscopy

Purified TMcin-G1905 supplemented with 0.1 mg/mL δ-toxin and *S. aureus* USA300 LAC cells were prepared and adjusted to an OD$_{600}$ = 5. Cells were treated for 30 min at 37 °C with 1 μM TMcin-G1905 or vehicle (see Cell-killing assays), after which paraformaldehyde was added to 4%.

Cells were collected and fixed with 2.5% glutaraldehyde in 0.1 M sodium cacodylate buffer. After three buffer washes, the cells were post-fixed with 1% osmium tetroxide, reduced with 0.8% potassium ferrocyanide in 0.1 M sodium cacodylate buffer for an hour. All subsequent steps were carried out in a Pelco Biowave microwave (Ted Pella, Inc., Redding, CA) at 250 Watts and under vacuum. The cell pellets were washed with distilled water between each of the following treatments. The cells were treated with 1% tannic acid, en bloc stained with (Uranyl acetate replacement) UAR, and dehydrated with graded ethanol series. After dehydration with 100% ethanol, the cells were infiltrated with Epon/Araldite resin without accelerator. On the following day, the cells were pelleted and resuspended in 100% resin. After two exchanges of 100% resin, the cell pellets were embedded in 100% resin with the accelerator. The cell pellets were polymerized in a 60-degree oven overnight. The embedded cells were sectioned using a Leica ultramicrotome (Leica Microsystem, Wetzlar, Germany) to collect 70–90 nm thin sections on a 200-mesh copper grid. The grids were imaged using a bottom-mount AMT camera on an 80 kV Hitachi 7800 (Hitachi High Technologies, Tokyo, Japan) transmission electron microscope.

## Carboxyfluorescein liposome leakage assay

1,2-Dimyristoyl-sn-glycero-3-phosphoglycerol (DMPG, Avanti #840445 P) and cardiolipin (Avanti #710332 P) were dissolved in chloroform and mixed at a 7:3 molar ratio. Chloroform was removed using a rotary evaporator and an overnight incubation under vacuum. Assay buffer was supplemented with 50 mM 5(6)-carboxyfluorescein (Thermo Scientific) and used to resuspend lipids to 10 mg/mL. The lipid suspension was sonicated in a 37 °C water bath and extruded at 37 °C using a 1 μm polycarbonate membrane. Free dye was removed by G-75 Sephadex (Cytiva) gel filtration. Liposomes were collected and used immediately. TMcin-G1905 or vehicle was prepared in assay buffer supplemented with 7 μg/mL δ-toxin. TMcin-G1905, vehicle, 2% Triton X-100, or buffer were mixed in with carboxyfluorescein-entrapped liposomes in equal

volume, and fluorescence was monitored in a Tecan Spark microplate reader (monochromators: excitation, $485 \pm 20$ nm and emission $520 \pm 20$ nm). The fluorescence in samples with Triton X-100 or buffer only was set to 100% and 0%, respectively.

## Planar lipid bilayer assays

TMcin-G1905 and a mock control were purified in parallel as described from G1905 $\Delta agr$ and G1905 $\Delta agr$ pCT545::pCURE. The corresponding fractions of the mock purification as compared with the peptide were collected and treated identically.

Planar bilayers were formed by the lipid monolayer opposition technique using diphytanoyl-phosphatidylcholine (DPhPC) (Avanti Polar Lipids) on a circular aperture in a Teflon partition dividing two 1.5 mL (cis and trans) compartments of the experimental chamber, as previously described[74]. Aqueous solutions consisted of assay buffer supplemented with 150 mM KCl in both cis and trans compartments. For selectivity experiments, the trans compartment was supplemented with 750 mM KCl. Current records were performed as described previously[74] using Axopatch 200B amplifier (Axon Instruments) in the voltage-clamp mode. Data were filtered by a low-pass 8-pole Butterworth filter (Model 900 Frequency Active Filter, Frequency Devices) at 15 kHz, digitized with a sampling frequency of 50 kHz, and analyzed using pClamp 10.7 software (Axon Instrument). For data analysis, a digital filtering using a 1 kHz low-pass Bessel filter was applied. Potential is defined as positive when it is greater in cis side. Purified peptide or the mock purification was diluted in 50% acetonitrile, 0.1% TFA. Either TMcin-G1905 to a concentration of 20 nM or an equal volume of mock purification was added to the cis chamber after planar lipid membrane was formed, and its conductance of less than 0.01 pS was verified.

## Polysucrose uptake

Purified TMcin-G1905 supplemented with 0.1 mg/mL δ-toxin and *S. aureus* USA300 LAC cells were prepared and adjusted to an $OD_{600} = 1$. Cells were treated for 1 h at 37 °C with 1 μM TMcin-G1905 or vehicle (see Cell-killing assays). FITC-dextran 40 kD (Sigma #FD40S) or FITC-polysucrose 40 kD (Chondrex #CFP40-100mg) was resuspended in assay buffer to 12 mg/mL and added to cells to a final concentration of 1.2 mg/mL, and were incubated at 4 °C for 20 h. For flow cytometry, cells were collected by centrifugation, resuspended in assay buffer, and aspirated into a FACS Celesta Cell Analyzer (BD). One million forward scattering events were analyzed for fluorescence using a 488 nm excitation laser and a long pass 505 nm and 530/30 nm emission filters. For confocal analysis, cells were stained with DAPI and Nile Red. Images were acquired on a ZEISS LSM800 Confocal microscope.

## Phylogenetic analyses

TmcA sequences were obtained by PSI-BLAST in October 2021 using the translated product of the G1905 *tmcA* ORF, aligned by MAFFT, then a tree was made with RAxML using the general time reversible (GTR) GAMMA model and the rapid bootstrapping option with 100 replicates all run in Geneious Prime (version 2020.2.4) (Biomatters, New Zealand). Tree figure was rendered using ggtree in R[75].

## AlphaFold2 analyses

AlphaFold2 modelling of the monomer sequence (AWFVVLLAAILVFA-TAIFAGLTIWCVVNQHGKFTGNWNWHIKGVSLDVECKR) was performed using the Colaboratory cloud service[76]. Multimer modelling was carried out on a local installation of ColabFold[76]. Initial modelling was carried out with stoichiometries ranging from 2-25 with the following parameters: --msa-mode mmseqs2_uniref_env --pair-mode unpaired_paired --pair-strategy greedy --num-recycle 20 --recycle-early-stop-tolerance 1.0. The best model ($n = 21$) was then used as a template for a subsequent round ($n = 18$–30) with the same parameters as above. Structural images were rendered using ChimeraX (University of

California, San Francisco). A disulfide bond was manually added between Cys25 γS and Cys50 γS in the monomer AlphaFold2 model. For the prediction of multimer structures, we used the overall model confidence score developed by DeepMind[77]. The score is a weighted average of $0.8 \times \text{ipTM} + 0.2 \times \text{pTM}$ metrics that are outputs of AF2 multimer prediction.

## Circular dichroism (CD) spectroscopy

To remove TFA, purified TMcin-G1905 was lyophilized and resuspended with 50% ACN to 20 μM. TMcin-G1905 and vehicle (50% ACN) were diluted 10-fold with 5 mM $KPO_4$, pH 7.0, 2.5 mM $MgSO_4$ to a concentration of 2 μM. Measurements were performed using a Jasco J810 Spectropolarimeter with a 2 mm pathlength and scanning from 190 to 240 nm at a step size of 1 nm and an integration time of 1 s. The baseline was subtracted using the buffer-matched diluted vehicle, and the signal was smoothed using the Savitsky-Golay filter (window = 11 nm, polynomial order = 3). Secondary structure analysis was performed by matching the spectrum to the SMP180 reference set[1] using CONTINLL[2] on DichroWeb[3].

## Molecular dynamics (MD) simulations

Atomistic systems of the monomeric TMcin peptide in solution (10 nm cubic box) and 21 copy pore model embedded in a 7:3 DMPG:cardiolipin bilayer (189 and 81 molecules, respectively; 13.9*13.9*8.6 nm rectangular box) were constructed using CHARMM-GUI[1] with the CHARMM36[2] force field, and simulated using GROMACS 2024[3] using the standard CHARMM-GUI scripts and run parameters for energy minimization, equilibration, and production. The charge-neutral monomer and water-filled pore systems contained 150 mM KCl, TIP3 water (10,334 and 37,744 molecules, respectively), and were simulated in triplicate at 310.15 K for 1 μs and 0.5 μs, respectively. Root mean square fluctuation (RMSF) and root mean square deviation (RMSD) were calculated relative to the initial AF2 model using GROMACS tools. Secondary structure analysis and molecular visualization were performed using DSSP[4,5] and VMD[6], respectively.

## Statistics and reproducibility

Statistical tests were performed and graphs were created using the R language and R Studio IDE. ANOVA was used to test for differences across groups, and the Tukey Honestly Significant Difference method was used to test for pairwise differences. No statistical method was used to predetermine sample size. No data were excluded from the analyses. The experiments were not randomized.

## Reporting summary

Further information on research design is available in the Nature Portfolio Reporting Summary linked to this article.

# Data availability

The TMcin-G1905 protein sequence data reported in this paper will appear in the UniProt Knowledgebase under the accession number C0HMD4. Source data used to generate graphs are provided with this paper. The AlphaFold2 and flow cytometry data files are available in the Digital Repository at the University of Maryland[78]. Material requests for plasmid and bacterial strains generated in this study are available upon request from Seth Dickey (sdickey@umd.edu). Source data are provided with this paper.

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

## Acknowledgements

The authors would like to thank Jan Maarten van Dijl (University Medical Center Groningen) for kindly providing the CT545 and W13 strains, Brian Martin (National Institute of Mental Health) for help with Edman degradation, Vinod Nair and Margery Smelkinson (National Institute of Allergy and Infectious Diseases, Research Technologies Branch) for help with TEM and confocal/deconvolution microscopy, respectively, and Kevin McIver (University of Maryland, College Park) for kindly providing *Streptococcus pyogenes*. This work was funded by the NIH Intramural Research Programs of the National Institute of Allergy and Infectious Diseases (project number ZIA AI000904 to M.O.), the *Eunice Kennedy Shriver* National Institute of Child Health and Human Development (project number ZIA HD000072 to S.M.B.), the National Institute of Allergy and Infectious Disease BCBB Support Services Contract (HHSN316201300006W/75N93022F00001 to Guidehouse, Inc., to M.G.), the Canadian Institutes of Health Research (to N.C.J.S and D.P.T), the Digital Research Alliance of Canada (to D.P.T), the Canada Research Chairs program (to D.P.T) and the University of Maryland startup funds (to S.W.D.).

## Author contributions

S.W.D. and M.O. designed the research; S.W.D., D.J.B., and A.V. performed molecular cloning and allelic exchange. S.W.D., D.J.B., and T.K.R. performed planar lipid bilayer experiments; S.M.B. provided guidance on PLB experiments; M.G. performed bioinformatic analyses; G.Y.C.C. raised antisera against the TMcin-G1905; S.W.D., D.J.B., L.J.W., A.C.L., and N.C.J.S. performed AlphaFold2 analyses; A.N.A. performed FITC-polysucrose uptake and CD experiments; E.A.C. and D.P.T. conducted MD simulations; S.W.D., D.J.B., and M.O. performed all other

research; S.W.D. and M.O. wrote the manuscript and prepared figures; all authors edited the manuscript.

## Competing interests

The authors declare no competing interests.
