## [Transparent Peer Review file · Nature Communications]

Antimicrobial peptide class that forms discrete β -barrel stable pores anchored by transmembrane helices

Corresponding Author: Dr Seth Dickey

Version 0:

Reviewer comments:

Reviewer #1

(Remarks to the Author)

Title: Antimicrobial peptide class that forms discrete beta-barrel stable pores anchored by transmembrane helices

The manuscript, titled "Antimicrobial peptide class that forms discrete beta-barrel stable pores anchored by transmembrane helices", is a significant contribution that shows an AMP class that achieves pore stability otherwise only found in protein toxins. While the manuscript is interesting and could contribute further, some areas could be improved, and we have provided some suggestions to enhance the manuscript's quality.

Suggestions

1. The discovery of TMxin is exciting and contributes directly to the AMP field. Nevertheless, structural evaluation is too preliminary and could be improved. Despite the authors utilizing alpha-fold2, which is efficient for proposing protein structures, the efficiency is not the same for short peptides. The proposed structure is a beta-sheet associated with two helices, which are usually found in defensins. However, the sequence does not have the cysteines in the correct position for a defensin structural scaffold and a unique disulfide bond in a strange position (between the helix and the beta-sheet). In this case, it seems that the structure proposed could be an artifact. To remove this doubt, I strongly suggest that the authors run an NMR and CD analysis. Moreover, once the authors have analyzed the monomers through structural analyses, alpha-fold could also help predict their interaction during multimer formation. In this case, Iptm values must be considered. Not less important, authors must also simulate the peptide in membrane and water environments by using molecular dynamics. This study will provide more reliability about the structure and the mechanism of action.
2. In Figure 6, it is unclear how the authors calculated the number of monomers to construct the multimer. Since the methods are not described, the authors must better explain the construction obtained for Figures 6B and 6G.
3. In supplementary Figure 4, the score calculated was 0.8 piTM. Please provide more details about that. Is this calculation performed between each monomer? Please clearly describe the methods. Moreover, how did authors know that such a structure was stable inside a membrane? Please provide a membrane simulation.
4. 3. Another suggestion is cryo-TEM to show pore formation and stabilization. This technology could significantly enhance the evidence of this novel mechanism of action, making the manuscript more convincing and impactful.

Reviewer #2

(Remarks to the Author)

Major comments

AMPs are generally defined as peptides being produced as part of the invertebrate, plant and animal innate defense against microorganisms. TMxins are produced by bacteria and it seems incorrect to refer to TMxins as AMPs. It would be more appropriate to refer to TMxins as bacteriocins throughout the text which would require that the manuscript is focused differently and that the entire manuscript including the introduction and discussion is carefully rewritten to provide adequate background information on bacteriocins and to discuss the findings in this perspectives. In this respect, what is known about the structure of other bacteriocins, nisin for example? And what is the main novelty of the manuscript?

Alternatively, the authors should explain very clearly why they consider TMxins to be AMPs instead of bacteriocins.

The presentation of the results is in places superficial with a lot of abbreviations and important background information not provided – detailed below.

Specific comments

Summary, nor very clear

Line 67 + 76: Combine to one paragraph and shorten.

Line 82, unclear why the term BGC is used here – so far these plasmid associated genes have not been assigned a function and may be functionally unrelated.

Line 102, PSMs not defined

Line 115-116: a repetition of Line 112-114?

Line 119, please give some more information in the text about how these homologs were identified, in what species they were detected (there is some information in Line 138, however, it would be more logical to have the information here), how similar they are, and if the gene(s) are always found on plasmids.

Line 126-37: convoluted as written, please rephrase to improve clarity.

Line 143-44: no experimentally proven function has been assigned to these putative genes and also it has not been validated that they function together as a BGC

Cell killing assays: “vehicle” is that the buffer? Is TMxin active against stationary phase cells? Please explain

Line 181 “did not lyse cells”, is not consistent with the images showing membrane damage, loss of turgor, and condensed DNA as observed for nisin -treated *S. aureus*?

Line 236-237: Unclear sentence “Nonetheless, our validation justifies a coarse structural interpretation, making TMxin the first nanoscale pore-forming AMP family to be supported with explicit structural models (18,60)” and as explained above I don't think such a conclusion is justified by the work presented.

Reviewer #3

(Remarks to the Author)

Dickey et al reveal a new class of bacteriocins which has been overlooked so far despite being found in many Firmicutes and Actinobacteria. The TMxins have a unique way of forming pores and appear to play important roles in several ecosystems including human microbiomes and infection.

I have a few questions that should be addressed in the revised manuscript:

1. It is interesting to note that the TMxin BCG is exclusively found in a specific *S. aureus* clone isolated from Buruli ulcers. Please explain the disease and its causative agents briefly in the introduction. What is known about this clonal lineage? Is it also found in other types of infection? Is it found in the human nose?
2. Does the TMxin act against *Mycobacterium ulcerans* or other mycobacteria? Are these ulcers polymicrobial wounds and do the authors expect TMxins to support fitness in a wound microbiome?
3. Did the authors analyze potential toxicity for human cells? This is a critical point as it is found to be produced in human infections. The term “cytotoxin” implies activity against human cells and may be misleading. If human cells are insensitive, what could be the reason?
4. Does the leader peptide resemble those of any other bacteriocins or is it unique?
5. Line 68: antimicrobial activity against which bacteria?
6. Is DsbA an intracellular or extracellular protein? Do the authors expect it to introduce SS bonds before or after TMxin secretion?
7. PSMs have been found to be released as components of membrane vesicles. Could the same be true for the hydrophobic TMxins?

Version 1:

Reviewer comments:

Reviewer #1

(Remarks to the Author)

Dear Dr. Dickey,

The newly added data, including molecular dynamics simulations of both the monomer and the 21-mer pore structure in a lipid bilayer, significantly strengthen the structural claims made in the manuscript. The inclusion of circular dichroism (CD) spectroscopy provides further support for the predicted secondary structure and helps validate the AlphaFold2 (AF2) model in the absence of high-resolution experimental data.

The clarification of the methodology used for pore stoichiometry prediction and the improved description of the AF2 scoring metrics, including the integration of ipTM and pTM scores, are welcome additions that enhance the rigor and transparency of the structural modeling.

While I still consider experimental structural validation methods like NMR or cryo-TEM to be ideal, I acknowledge the technical challenges associated with the peptide's hydrophobicity, low yield, and solubility. The authors have demonstrated due diligence in attempting to address these limitations and have clearly articulated the constraints. The decision to proceed with the available data, which is now more robust and comprehensive, seems justified.

In summary, the revised manuscript is substantially improved and now presents a compelling case for the existence of a novel AMP pore-forming mechanism, supported by a combination of computational and biophysical data.

I have no further major concerns, and I support the publication of this manuscript in its current revised form.

Reviewer #2

(Remarks to the Author)

I enjoyed reading the revised manuscript and have no further comments

Reviewer #3

(Remarks to the Author)

All my concerns were appropriately addressed.

REVIEWER COMMENTS

Reviewer #1 (Remarks to the Author):

Title: Antimicrobial peptide class that forms discrete beta-barrel stable pores anchored by transmembrane helices

The manuscript, titled “Antimicrobial peptide class that forms discrete beta-barrel stable pores anchored by transmembrane helices”, is a significant contribution that shows an AMP class that achieves pore stability otherwise only found in protein toxins. While the manuscript is interesting and could contribute further, some areas could be improved, and we have provided some suggestions to enhance the manuscript's quality.

We thank the reviewer for their interest and careful consideration of our manuscript.

Suggestions

1. The discovery of TMxin is exciting and contributes directly to the AMP field. Nevertheless, structural evaluation is too preliminary and could be improved. Despite the authors utilizing alpha-fold2, which is efficient for proposing protein structures, the efficiency is not the same for short peptides. The proposed structure is a beta-sheet associated with two helices, which are usually found in defensins. However, the sequence does not have the cysteines in the correct position for a defensin structural scaffold and a unique disulfide bond in a strange position (between the helix and the beta-sheet). In this case, it seems that the structure proposed could be an artifact. To remove this doubt, I strongly suggest that the authors run an NMR and CD analysis. Moreover, once the authors have analyzed the monomers through structural analyses, alpha-fold could also help predict their interaction during multimer formation. In this case, Iptm values must be considered. Not less important, authors must also simulate the peptide in membrane and water environments by using molecular dynamics. This study will provide more reliability about the structure and the mechanism of action.

We are encouraged by the enthusiasm of the reviewer in that our findings directly impact the AMP field.

To improve the structural evaluation of the AlphaFold2 (AF2) predicted structures, we reached out to experts (new co-authors) in conducting molecular dynamics simulations (MD) who ran MD simulations of the TMcin-G1905 monomer in solution and the predicted 21-mer oligomeric pore in a lipid bilayer environment that models the target Gram-positive cellular membrane (Figs. 6h-l, Supplementary Figure 5, Results lines 223-239). Both the TMcin monomer and the oligomeric pore retained the fold predicted by AF2 throughout the μ s-long MD simulation in solution and the 0.5 μ s-long simulation in a lipid bilayer. A subset of residues within the monomer structure exhibited substantial fluctuation, however, this was expected given the small size of the peptide. In contrast, all residues within the 21-mer pore structure exhibited little fluctuation, which is likely due to the non-polar lipid environment and the extensive interactions between β -strands of the protomers that form the β -barrel.

We also analyzed TMcin-G1905 in solution using circular dichroism spectroscopy (CD; Figure 6m, Supplementary Table 3, Results lines 240-245). The CD results revealed both α -helical and β -sheet secondary structures, which supports the AF2 models and the MD simulations. Interestingly, the CD spectrum also demonstrated a substantial fraction of disordered residues, which is consistent with the monomer MD simulation that showed fluctuations of residues within the β strands and the N-terminus of the α -helix.

AF2 has been benchmarked with 588 peptides using NMR solution structures (ref.56, McDonald et al., 2023). Overall AF2 performed well, especially in predicting secondary structure of mixed secondary structure peptides and for α -helical transmembrane segments. We have revised the discussion (lines 263-265) accordingly and included this reference.

We apologize for unintentionally omitting the reference from DeepMind in establishing the model confidence score. This score is a weighted average of the ipTM and pTM scores ($0.8 \cdot \text{ipTM} + 0.2 \cdot \text{pTM}$) and thereby accounts for interfacial interactions. We have added the reference to the Methods section (line 579). The formula for the model confidence score is included in the Methods (lines 578-579) and in the figure legend of Supplementary Figure 4.

Taken together, the CD and MD analyses along with the referenced AF2 benchmarking boost confidence in our reported AF2 predicted models. Moreover, the models are consistent with the behavior of TMcin-G1905 in biophysical and cellular assays reported throughout our manuscript.

We also note that NMR analyses at this time are not feasible given the limited yield (approximately 100 μg), the challenges in synthesizing a hydrophobic 52-residue peptide, and the solubility (approximately 0.1 mg/mL; 20 μM) of the TMcin peptide even when supplemented with organic solvent. We have invested substantial resources in increasing yields, however, our efforts have yet to provide sufficient material for NMR or cryo-TEM analyses.

2. In Figure 6, it is unclear how the authors calculated the number of monomers to construct the multimer. Since the methods are not described, the authors must better explain the construction obtained for Figures 6B and 6G.

We apologize for not making our approach clearer in the text. We have modified the text of the Results (line 211-213) to read, “Without any a-priori insight into the TMcin pore stoichiometry, we adopted an agnostic approach and queried AF2 using all n-mers between 2 and 30 to predict pore structures.”

3. In supplementary Figure 4, the score calculated was 0.8 piTM. Please provide more details about that. Is this calculation performed between each monomer? Please clearly describe the methods. Moreover, how did authors know that such a structure was stable inside a membrane? Please provide a membrane simulation.

As discussed above, DeepMind, the creators of AF2, established an overall model confidence score for assessing predicted multimer structures, such as the oligomeric pore models shown in Supplementary Figure 4. The score is a weighted average of $0.8 \cdot \text{piTM} + 0.2 \cdot \text{pTM}$ metrics that are outputs of AF2 multimer prediction. pTM is the predicted template modeling score and is analogous to the widely used TM score that compares the similarity of two structures. AF2 pTMs accurately represent TMs (ref.36, Jumper et al.,

2021). ipTM is the interface pTM score used for multimer predictions that assesses the confidence of interactions between residues on different chains. We have provided additional explanation in the Methods section (lines 577-579).

Following the reviewer's suggestion, we have added MD simulations in the revised manuscript of the 21-mer in a lipid bilayer that models the Gram-positive cell membrane (Fig. 6i-l, Supplementary Figure 5c-d). Notably, the pore structure exhibits minimal fluctuation (<0.2 nm for all residues), suggesting that the proposed pore structure is stable within a membrane environment. We have revised the Results (lines 223-239) to include the new data.

4. 3. Another suggestion is cryo-TEM to show pore formation and stabilization. This technology could significantly enhance the evidence of this novel mechanism of action, making the manuscript more convincing and impactful.

We thank the reviewer for this suggestion, and we expended substantial efforts in pursuing structural evaluation using cryo-TEM to better understand the TMcin pore. However, completing these analyses will take a substantial amount of more time and effort due to the limitations of working with an extremely hydrophobic peptide as discussed above and the extensive efforts needed to optimize cryo-TEM analysis of the TMcin pore within a membrane environment. We judge that novelty of the TMcin pore with the evidence contained within the manuscript, both in predicted structures and in biophysical and cell-based experiments that support a model in which TMcin forms large pores in a lipid bilayer and in cell membranes, merits communication of our findings at this time with the scientific community.

Reviewer #2 (Remarks to the Author):

Major comments

AMPs are generally defined as peptides being produced as part of the invertebrate, plant and animal innate defense against microorganisms. TMxins are produced by bacteria and it seems incorrect to refer to TMxins as AMPs. It would be more appropriate to refer to TMxins as bacteriocins throughout the text which would require that the manuscript is focused differently and that the entire manuscript including the introduction and discussion is carefully rewritten to provide adequate background information on bacteriocins and to discuss the findings in this perspectives. In this respect, what is known about the structure of other bacteriocins, nisin for example? And what is the main novelty of the manuscript?

Alternatively, the authors should explain very clearly why they consider TMxins to be AMPs instead of bacteriocins.

We thank the reviewer for their time in reading our manuscript and appreciate the suggestion of more precisely referring to TMxin as a bacteriocin. We agree that in many labs working on bacteriocins and host-microbe interactions, the term "antimicrobial peptide" (AMP) is often used to contrast bacteriocins, which are antimicrobially active substances produced by bacteria. However, by definition, according to many references, and we believe also the general understanding in the AMP community, bacteriocins are

a subclass of AMPs, and thus our newly discovered antimicrobially active substance is both a bacteriocin and an AMP.

However, we understand the concern of the reviewer and therefore, to increase the specificity of our naming, we have renamed TMxin to TMcin (for transmembrane-helix containing bacteriocin) and the genes on the putative BGC from *tmx* to *tmc* (e.g., *tmxA* to *tmcA*). In addition, we have added discussion about bacteriocins and AMPs throughout the text (lines 26, 46, 144, & 257). Furthermore, TMcin can be classified as a RiPP (ribosomally produced and post-translationally modified peptide), which helps to contrast it to non-ribosomal peptides that can also be produced by bacteria and can act as AMPs. We have revised the text to include TMcin as a RiPP (lines 155-157).

The presentation of the results is in places superficial with a lot of abbreviations and important background information not provided – detailed below.

Specific comments

Summary, nor very clear

Line 67 + 76: Combine to one paragraph and shorten.

We have combined into one paragraph and shortened to improve clarity.

Line 82, unclear why the term BGC is used here – so far these plasmid associated genes have not been assigned a function and may be functionally unrelated.

In this context, we used the term BGC to refer to other known characterized biosynthetic gene clusters. We also revised the text to more carefully introduce the TMcin BGC as putative when appropriate.

Line 102, PSMs not defined

We have revised the text to define PSMs at the first mention (lines 104-105).

Line 115-16: a repetition of Line 112-114?

We revised the text to remove the apparent repetition (lines 115-118).

Line 119, please give some more information in the text about how these homologs were identified, in what species they were detected (there is some information in Line 138, however, it would be more logical to have the information here), how similar they are, and if the gene(s) are always found on plasmids.

Our focus at this point in the text (original manuscript, line 119) was to establish the conservation of the transmembrane helix and we feel that discussing the species in which the homologs were detected distracts from this focus. We did add, “see further below on details regarding homologous systems” to a parenthetical to direct readers interested in TMcin homology (lines 121-122).

Line 126-37: convoluted as written, please rephrase to improve clarity.

We have rephrased to improve the clarity (lines 129-140).

Line 143-44: no experimentally proven function has been assigned to these putative genes and also it has not been validated that they function together as a BGC

We agree with the reviewer that the function of individual gene products encoded on the putative TMcin BGC have not been validated, even though Figure 1c shows that the constellation of genes does confer the ability to produce TMcin-G1905 and that the homology of individual genes matches the processing required to produce mature TMcin-G1905. Nonetheless, a comprehensive characterization of the TMcin BGC is out of the scope of this current study and awaits further investigation.

Cell killing assays: “vehicle” is that the buffer? Is TMxin active against stationary phase cells?

Please explain

As described in the methods section under “Cell-killing assays”, vehicle is 50% acetonitrile (ACN) and 0.1% trifluoroacetic acid (TFA) that we used to solubilize purified TMcin at a stock concentration of 20 μ M (lines 349-377 & 401). It is important to note that the corresponding working concentrations of ACN and TFA are lower after mixing with assay buffer and δ -toxin. For example, at 1 μ M TMcin-G1905, the concentrations are 2.5% ACN and 0.005% TFA.

Line 181 “did not lyse cells”, is not consistent with the images showing membrane damage, loss of turgor, and condensed DNA as observed for nisin -treated *S. aureus*?

Our intended meaning of lysis referred to a complete breakdown of the cell membrane and cell wall that would lead to release of all intracellular content including DNA. We thank the reviewer for pointing out “lysis” is not precisely defined and we have revised the text to, “did not lead to the dissolution of the cell membrane” to improve clarity (line 187-188).

Line 236-237: Unclear sentence “Nonetheless, our validation justifies a coarse structural interpretation, making TMxin the first nanoscale pore-forming AMP family to be supported with explicit structural models (18,60)” and as explained above I don’t think such a conclusion is justified by the work presented.

We have revised the discussion and deleted the ambiguous statement, “course structural interpretation”. In addition to improving our structural evaluation with MD simulations and CD spectroscopy discussed above, we also revised the discussion to limit claims of TMcin being the first nanoscale pore-forming AMP supported with explicit structural models.

Reviewer #3 (Remarks to the Author):

Dickey et al reveal a new class of bacteriocins which has been overlooked so far despite being found in many Firmicutes and Actinobacteria. The TMxins have a unique way of forming pores and appear play important roles in several ecosystems including human microbiomes and infection.

I have a few questions that should be addressed in the revised manuscript:

1. It is interesting to note that the TMxin BGC is exclusively found in a specific *S. aureus* clone isolated from Buruli ulcers. Please explain the disease and its causative agents briefly in the

introduction. What is known about this clonal lineage? Is it also found in other types of infection? Is it found in the human nose?

The reviewer raises an important question: what are the biological roles of TMcin and its associated antimicrobial activity? We have added this question to the discussion (lines 283-286) to highlight the need for future work in this area.

Other sequenced *S. aureus* strains unrelated to Buruli ulcers encode for the TMcin BGC. In addition, many of the other Gram-positive bacteria that encode for homologous TMcin BGCs are not known to colonize or cause infections in humans. Therefore, the biological roles of TMcin are likely myriad and dependent on the environmental conditions and the presence of other competitors. We focused this initial manuscript to report on their discovery and to characterize the mechanism of antimicrobial activity. In addition, we demonstrated as a proof of principle that TMcin confers a competitive advantage in co-culture (Fig. 2). Thereby, this manuscript sets the foundation for future efforts to understand specific biological roles that TMcin plays.

2. Does the TMcin act against Mycobacterium ulcerans or other mycobacteria? Are these ulcers polymicrobial wounds and do the authors expect TMcins to support fitness in a wound microbiome?

The reviewer again raises important biological questions about TMcin and how it has the potential to shape the microbiome of an environment. We do note that TMcin-producing *S. aureus* may not have been selected for the potential killing of *M. ulcerans* by TMcin because these *S. aureus* strains were isolated as secondary infections after antibiotic treatment cleared the original *M. ulcerans* infections. Nonetheless, others have described Buruli ulcer wounds as polymicrobial and it is reasonable, though speculative at this time, to expect that TMcin increases the fitness of *S. aureus* in the wound environment.

3. Did the authors analyze potential toxicity for human cells? This is a critical point as it is found to be produced in human infections. The term “cytotoxin” implies activity against human cells and may be misleading. If human cells are insensitive, what could be the reason?

The reviewer raises an important consideration when evaluating TMcin as a potential therapeutic. Activity against human cells is often observed among bacteriocins and other AMPs. However, our focus was to describe our discovery of this family and to provide a mechanistic understanding of antimicrobial activity and the evaluation of toxicity against human cells is out of the scope of this study. To avoid misleading readers and implying TMcin has toxin activity against human cells, we have renamed TMxin to TMcin (transmembrane helix containing bacteriocin), eliminating “cytotoxin” from the name.

4. Does the leader peptide resemble those of any other bacteriocins or is it unique?

The leader protein sequence of TMcin-G1905 does not exhibit homology to those of other bacteriocins and we have included this in the Results of the revised manuscript (lines 143-143).

5. Line 68: antimicrobial activity against which bacteria?

In the legend for Fig. 1a, we describe that the activity is against *Micrococcus luteus*. For clarity, we did not amend the text in the Results.

6. Is DsbA an intracellular or extracellular protein? Do the author expect it to introduce SS bonds before or after TMxin secretion?

Although out of scope of this first publication, this excellent question is relevant to the biosynthesis of the TMcin peptide and functional analysis of the TMcin BGC gene products. We anticipate future studies that focus on TMcin biosynthesis.

7. PSMs have been found to be released as components of membrane vesicles. Could the same be true for the hydrophobic TMxins?

The reviewer raises another insightful question. However, in the case of wild-type G1905, the elution of TMcin-G1905 from culture filtrates using a Superdex 200 gel filtration column (Fig. 4e) is inconsistent with TMcin-G1905 existing as a component of membrane vesicles. Although it is possible that high-molecular weight TMcin from the G1905 Δagr associates with membrane vesicles in the absence of PSMs, this potential phenomenon is out of the scope of the current study.

REVIEWERS' COMMENTS

Reviewer #1 (Remarks to the Author)

Dear Dr. Dickey,

The newly added data, including molecular dynamics simulations of both the monomer and the 21-mer pore structure in a lipid bilayer, significantly strengthen the structural claims made in the manuscript. The inclusion of circular dichroism (CD) spectroscopy provides further support for the predicted secondary structure and helps validate the AlphaFold2 (AF2) model in the absence of high-resolution experimental data.

The clarification of the methodology used for pore stoichiometry prediction and the improved description of the AF2 scoring metrics, including the integration of ipTM and pTM scores, are welcome additions that enhance the rigor and transparency of the structural modeling.

We thank the reviewer for their thoughtful feedback. We are pleased that the reviewer finds that the newly added data support the claims of the manuscript. We also appreciate the reviewer's recognition of our efforts towards transparency and rigor in the structural modeling.

While I still consider experimental structural validation methods like NMR or cryo-TEM to be ideal, I acknowledge the technical challenges associated with the peptide's hydrophobicity, low yield, and solubility. The authors have demonstrated due diligence in attempting to address these limitations and have clearly articulated the constraints. The decision to proceed with the available data, which is now more robust and comprehensive, seems justified.

In summary, the revised manuscript is substantially improved and now presents a compelling case for the existence of a novel AMP pore-forming mechanism, supported by a combination of computational and biophysical data.

I have no further major concerns, and I support the publication of this manuscript in its current revised form.

We appreciate the reviewer's recognition of the technical challenges associated with experimental structural validation for this peptide and we are grateful for their acknowledgment of the steps we have taken to address these limitations. We are especially encouraged by the assessment that the data now present a compelling case for a novel AMP pore-forming mechanism. We appreciate their endorsement of the manuscript in its current form.

Reviewer #2 (Remarks to the Author)

I enjoyed reading the revised manuscript and have no further comments

We thank the reviewer for their time and consideration. We are thoroughly pleased that the reviewer enjoyed reading our revised manuscript.

Reviewer #3 (Remarks to the Author)

All my concerns were appropriately addressed.

We thank the reviewer for their constructive feedback and are appreciative that all concerns were addressed.